# Genome-wide discovery and integrative genomic characterization of insulin resistance loci using serum triglycerides to HDL-cholesterol ratio as a proxy

Natalie DeForest[1], Yuqi Wang[1], Zhiyi Zhu[1], Jacqueline S. Dron [2,3], Ryan Koesterer[3], Pradeep Natarajan[2,4], Jason Flannick [3,5], Tiffany Amariuta[6,7], Gina M. Peloso [8] & Amit R. Majithia [1]✉

Insulin resistance causes multiple epidemic metabolic diseases, including type 2 diabetes, cardiovascular disease, and fatty liver, but is not routinely measured in epidemiological studies. To discover novel insulin resistance genes in the general population, we conducted genome-wide association studies in 382,129 individuals for triglyceride to HDL-cholesterol ratio (TG/HDL), a surrogate marker of insulin resistance calculable from commonly measured serum lipid profiles. We identified 251 independent loci, of which 62 were more strongly associated with TG/HDL compared to TG or HDL alone, suggesting them as insulin resistance loci. Candidate causal genes at these loci were prioritized by fine mapping with directions-of-effect and tissue specificity annotated through analysis of protein coding and expression quantitative trait variation. Directions-of-effect were corroborated in an independent cohort of individuals with directly measured insulin resistance. We highlight two phospholipase encoding genes, *PLA2G12A* and *PLA2G6*, which liberate arachidonic acid and improve insulin sensitivity, and *VGLL3*, a transcriptional co-factor that increases insulin resistance partially through enhanced adiposity. Finally, we implicate the anti-apoptotic gene *TNFAIP8* as a sex-dimorphic insulin resistance factor, which acts by increasing visceral adiposity, specifically in females. In summary, our study identifies several candidate modulators of insulin resistance that have the potential to serve as biomarkers and pharmacological targets.

Insulin resistance is a major cause of multiple chronic and epidemic diseases, including type 2 diabetes (T2D), non-alcoholic fatty liver disease (NAFLD), cardiovascular disease (CVD), and cancer[1,2]. Environment, lifestyle and genetics[3] contribute to the pathogenesis of insulin resistance, with heritability estimates for genetic factors ranging from 24 to 73%[4,5]. Family-based studies have implicated several insulin resistance genes[6–8], but modern genome-wide association studies (GWAS) accommodate larger sample sizes and enable the

[1]Division of Endocrinology, Department of Medicine, University of California San Diego, La Jolla, CA, USA. [2]Center for Genomic Medicine and Cardiovascular Research Center, Massachusetts General Hospital, Boston, MA, USA. [3]Programs in Medical and Population Genetics, Broad Institute of MIT and Harvard, Cambridge, MA, USA. [4]Department of Medicine, Harvard Medical School, Boston, MA, USA. [5]Department of Pediatrics, Boston Children's Hospital, Boston, MA, USA. [6]Halıcıoğlu Data Science Institute, University of California San Diego, La Jolla, CA, USA. [7]Division of Biomedical Informatics, Department of Medicine, University of California San Diego, La Jolla, CA, USA. [8]Department of Biostatistics, Boston University School of Public Health, Boston, MA, USA. ✉e-mail: amajithia@health.ucsd.edu

identification of genetic factors contributing to insulin resistance in the general population[9]. GWAS for insulin resistance, however, are challenging to perform in large population cohorts as the protocols for directly measuring whole-body insulin sensitivity, such as the hyperinsulinemic-euglycemic clamp[10], are difficult to perform at scale. To circumvent this limitation, GWAS have been conducted using surrogate markers of insulin resistance such as glycemic traits[11,12] and anthropomorphic measurements[13], which have successfully identified numerous risk loci. Larger sample sizes have further enabled sex-stratified analysis and the discovery of sex-dimorphic risk loci[13,14]. These studies have revealed valuable insights, but each surrogate marker only captures a portion of insulin resistance pathophysiology. For instance, GWAS for glycemic traits have identified genes related to hepatic glucose metabolism, and findings from anthropomorphic trait GWAS have largely implicated adipose tissue insulin resistance. The variation in loci/genes identified by these studies suggests that genetic mapping of other complementary insulin resistance biomarkers would deepen our knowledge of genetic factors modulating insulin resistance.

The triglyceride (TG) to HDL-cholesterol (HDL) ratio (TG/HDL) is a widely available surrogate marker of insulin resistance that is calculable from commonly measured clinical lipid profiles. It is highly correlated with hyperinsulinemic-euglycemic clamp based measurements[15] across multiple ethnicities[16,17] and weight categories[18,19], but can also capture information about lipid metabolism similar to individual TG or HDL measurements.

Here, to characterize the genetic basis of TG/HDL and systematically identify novel insulin resistance genes, we conducted a GWAS for TG/HDL in 382,129 UK Biobank (UKBB) participants and prioritized target genes at top genomic risk loci through statistical fine-mapping and combined single nucleotide polymorphisms (SNP)-to-gene linking strategies. Prioritized genes were further interrogated in multiple orthogonal genetic and genomic analyses to determine directional effects and tissue specificity.

## Results

### GWAS for TG/HDL in 382,129 UK Biobank participants identifies 251 associated loci

To confirm the relationship between TG/HDL and insulin resistance measurements, we compared the TG/HDL value concurrently obtained from a previous study of 45 individuals undergoing hyperinsulinemic-euglycemic clamp, the gold standard of insulin sensitivity quantification[20] (Supplementary Fig. 1A). We observed a strong and significantly negative phenotypic correlation between TG/HDL and clamp-based glucose disposal rate (Rd) ($\rho = -0.48$, $p = 0.003$, Spearman rank correlation) as well as with other commonly used surrogate markers such as fasting insulin (FI) ($\rho = 0.42$, $p = 0.009$).

Aiming to identify associated loci for TG/HDL, we conducted a GWAS on serum TG/HDL levels in 382,129 individuals from the UK Biobank who had both TG and HDL measurements available. Median age of the included individuals was 58 years, median BMI was 26.7, 54% were female, and 93% were of European ancestry. After quality control and filtering based on the Global Lipids Genetics Consortium standards[21,22] (Supplementary Fig. 1B), approximately 12 million SNPs were tested for association with TG/HDL using whole-genome linear regression as instantiated in REGENIE[23], including the first 20 principle components of genetic ancestry as covariates. The raw SNP associations were clustered into 251 TG/HDL loci using the FUMA platform[24] (Supplementary Data 1, Supplementary Fig. 2A). The genetic architecture of the 251 identified TG/HDL loci was consistent with a highly polygenic trait with the majority of identified loci tagged by common SNPs (223/251 lead SNPs with minor allele frequency (MAF) > 0.05) of relatively smaller effect sizes and fewer low-frequency variants (MAF < 0.01) with effect sizes 2–3x larger than those of lead common variants (Fig. 1a). Known insulin resistance signals from prior GWAS were re-

identified including *GCKR*[12], *IRS1*[25], *PPARG*[26], *FAM13A*[27], *ANGPTL4*[28], and *LPL*[29,30]. Lipid-associated loci for TG and HDL alone were also re-identified, including *MLXIPL*, *FADS2*, *CETP*, and *PLTP*[22]. Overlapping the lead SNPs marking these loci with tissue-specific regulatory elements (as instantiated in GREGOR[31]), we found enrichment in adipose tissue, liver, pancreatic islets, and skeletal muscle (Supplementary Fig. 3), all of which have been implicated in insulin sensitivity and insulin action[32].

Given the known sexual dimorphism in insulin resistance and metabolic disease risk[33], we also conducted a sex-stratified GWAS for TG/HDL in the same UKBB cohort (176,117 males vs. 206,012 females, Supplementary Fig. 2B, Supplementary Data 2) assessing each of the 251 genomic risk loci from the sex-combined TG/HDL GWAS ("Methods", Supplementary Data 1) for evidence of sexually dimorphic signals. Of the 251 loci, 17 showed significant evidence of sex-dimorphic genetic association ($p < 0.05/251$ t-test, Fig. 1a, Supplementary Data 1), with six of these having a stronger association in females (positive t-statistic). Among these were known sex-dimorphic insulin resistance genes *KLF14*[34] and *RSPO3*[35].

We attempted to corroborate these associations in the Mass General Biobank (MGBB, $n = 37,545$)[36], an independent dataset from the UKBB. Given the MGBB cohort was ten percent the size of the UKBB, we did not attempt to corroborate individual loci for lack of statistical power (only 11 loci had greater than 80% power for replication, Supplementary Data 1). Instead, we compared all loci in which UKBB discovery analysis that had proxy SNPs available in MGBB ($n = 240$), finding 40 loci robustly replicated ($p$-value < 0.05/240), 130 loci nominally replicated ($p$-value < 0.05), and 226 loci (94%) had the same direction of effect between UKBB and MGBB (Supplementary Data 1).

Having observed a phenotypic correlation between TG/HDL, clamp-based glucose infusion rate, and surrogate markers of insulin resistance (Supplementary Fig. 1A), we sought to quantify the degree to which TG/HDL, other surrogate insulin resistance markers, glycemic traits, and metabolic disease outcomes shared similar genetic influences. We computed the genetic correlations[37] between our TG/HDL GWAS and previous association studies of Homeostatic Model Assessment for Insulin Resistance (HOMA-IR)[12], Homeostatic Model Assessment for Beta Cell Function (HOMA-B)[38], fasting insulin (FI)[14], fasting glucose (FG)[14], waist-hip ratio adjusted for BMI (WHR)[13], 2 hour glucose (2 h glucose)[11], glycated hemoglobin (HbA1c)[11], Modified Stumvoll Insulin Sensitivity Index (ISI)[39] as well as metabolic disease sequelae including NAFLD[40], T2D[41], CVD[42] (Fig. 1b, Supplementary Data 3). Positive genetic correlations were observed between TG/HDL and metabolic disease outcomes: T2D ($r_g = 0.51$, $p = 1.9 \times 10^{-36}$), CVD ($r_g = 0.31$, $p = 2.8 \times 10^{-28}$), and NAFLD ($r_g = 0.74$, $p = 0.001$).

When considering glycemic traits and insulin secretion, the magnitude of genetic correlation between TG/HDL and HbA1c ($r_g = 0.12$) or 2 h glucose ($r_g = 0.23$) are comparable to FG ($r_g = 0.17$) and smaller than genetic correlation between TG/HDL and FI ($r_g = 0.49$), HOMA-IR ($r_g = 0.49$) or ISI ($r_g = -0.47$). Even though HOMA-B and HOMA-IR are calculated from the same fasting measurements, HOMA-IR ($r_g = 0.49$, $p = 4.11 \times 10^{-10}$) had a stronger and more significant correlation with TG/HDL than HOMA-B ($r_g = 0.41$, $p = 4.78 \times 10^{-7}$). Overall, these analyses show that TG/HDL is more genetically correlated with measures of insulin resistance than glycemia or insulin secretion.

### Prioritizing "boosted" TG/HDL associations to identify insulin resistance loci

Given that TG/HDL as a surrogate measure of insulin resistance is derived from serum TG and HDL, we next sought to identify signals that were "boosted" for association with TG/HDL versus TG or HDL alone under the model that boosted loci would be more likely to contain genes related to insulin resistance than lipid regulation. Using the lead SNP of each TG/HDL risk locus, we computed a "boost score" (TG/HDL_BS) utilizing the association $p$-values for TG/HDL, TG, and HDL

for that SNP in the UK Biobank (computed from the same samples with identical QC and analysis; "Methods"). Of the 251 loci identified in our study, 109 (43%) had boost score > 0, indicating increased significance in GWAS for TG/HDL compared to TG or HDL alone (Fig. 2a, red line). Focusing on known loci, we highlight that the *LPL* locus met the threshold for genome-wide significance for TG, HDL, and TG/HDL and had the highest boost score (TG/HDL$_{BS}$ = 50.4; TG/HDL *p*-value < $10^{-411}$, TG *p*-value < $10^{-324}$, HDL *p*-value = $10^{-362}$). These results are highly concordant with the well-characterized role of *LPL* in regulating both insulin sensitivity and serum lipid levels[43]. Other previously known and recently characterized insulin resistance loci including *IRS1*[25] (TG/HDL$_{BS}$ = 2.1), *PPARG*[26] (4.4), *RSPO3*[35] (6.0), *KLF14*[34] (12.2), and *COBLL1*[44] (9.0) demonstrated highly positive boost values as well (Fig. 2a). On the other hand, genes known to regulate lipids without relation to insulin sensitivity showed strongly negative boost scores. For example, the locus containing *CETP* had the lowest boost score (TG/HDL$_{BS}$ = −1539.8) driven by its substantially stronger association

with HDL ($p < 10^{-1952}$) compared to TG/HDL ($p < 10^{-412}$); this is consistent with the known role of *CETP* as a lipid transfer protein responsible for transferring cholesterol esters from HDL to other lipoproteins[45]. *ANGPTL3* also had a negative boost score (TG/HDL$_{BS}$ = −166.5) due to a stronger association with TG ($p < 10^{-309}$) compared to TG/HDL ($p < 10^{-143}$), as would be expected from its role in inhibiting TG hydrolysis through the inhibition of lipoprotein lipase[46]. Notably, the boost value does not perfectly segregate insulin resistance from lipid loci; for example, *APOB*, a known lipoprotein structural component, had a strongly positive boost score (TG/HDL$_{BS}$ = 5.5). We proceeded to rank the 251 associated loci according to their boost score and focused further investigation on the top quartile of loci ($n$ = 62, TG/HDL$_{BS}$ > 1.5, Supplementary Data 1).

Given the shared genetic basis of TG/HDL, other surrogate insulin resistance biomarkers, and metabolic disease outcomes, we performed a locus level comparison with previous biomarker/metabolic disease association studies to survey which genetic signals were

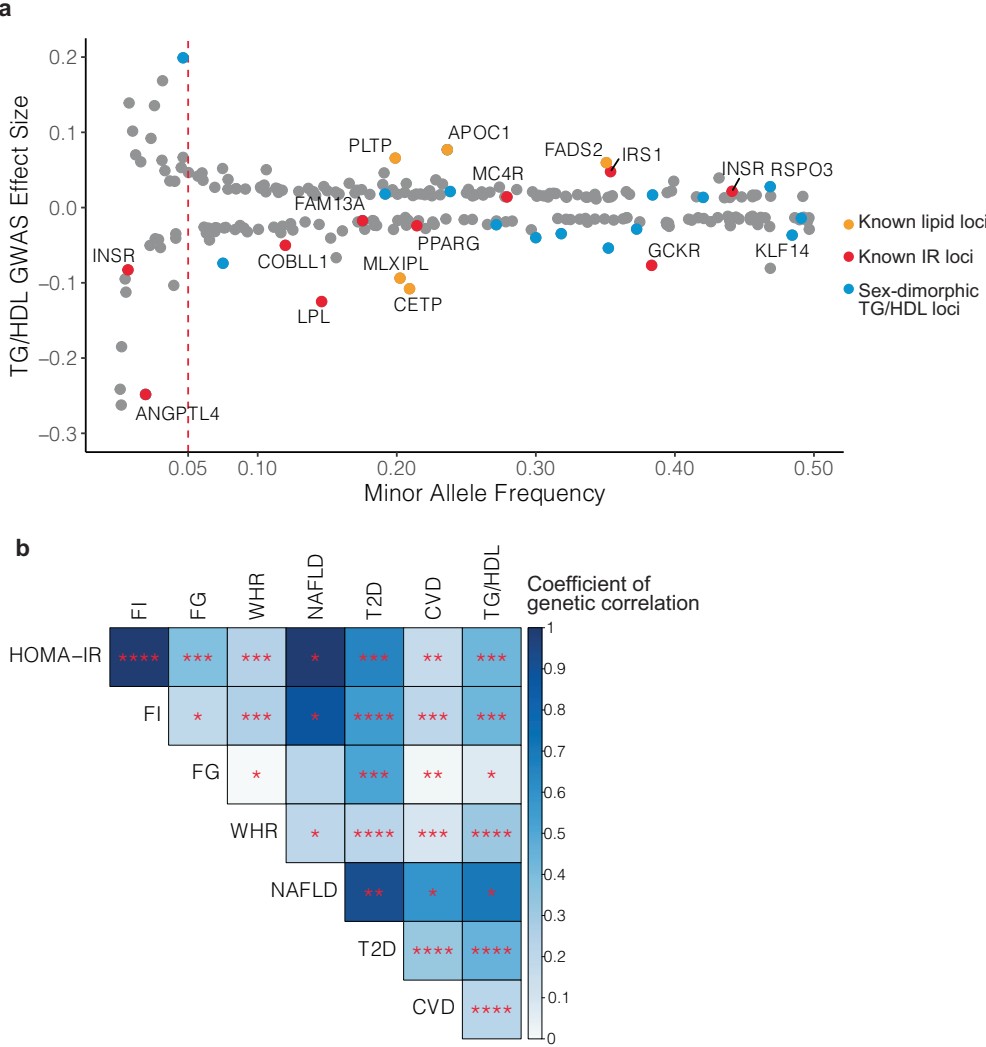

**Fig. 1 | GWAS for TG/HDL in 382,129 UK Biobank participants identifies 251 associated loci. a** Genetic architecture of 251 TG/HDL associated loci identified in the UK Biobank ($n$ = 382,129 participants). Shown are the effect sizes of the lead SNPs plotted according to minor allele frequency. Previously identified lipid and insulin resistance (IR) loci are colored orange and red, respectively. Loci with sex-dimorphic association signals are highlighted in blue ($n$ = 17/251). **b** Genetic correlations of TG/HDL with other surrogate measures of insulin resistance: fasting insulin (FI), fasting glucose (FG), waist-hip ratio (WHR), and metabolic diseases:

non-alcoholic fatty liver disease (NAFLD), type 2 diabetes (T2D), and cardiovascular disease (CVD). LD Score regression was used to estimate the *p*-values and the coefficients of genetic correlation. Color intensity represents the magnitude of the coefficient of genetic correlation, and statistical significance is indicated in the following bins: ****$p < $ 1e-20, ***$p < $ 1e-5, **$p < $ 0.001, *$p < $ 0.05. TG/HDL shows a strong positive genetic correlation with surrogate markers of insulin resistance and metabolic diseases.

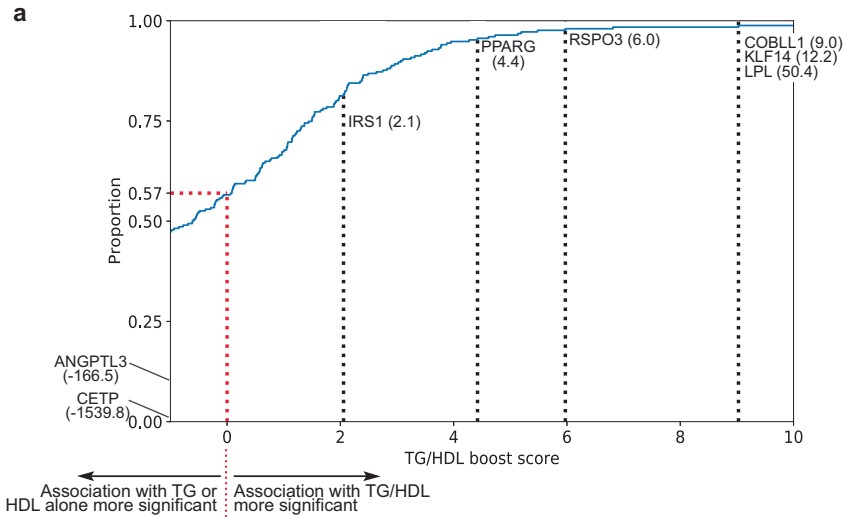

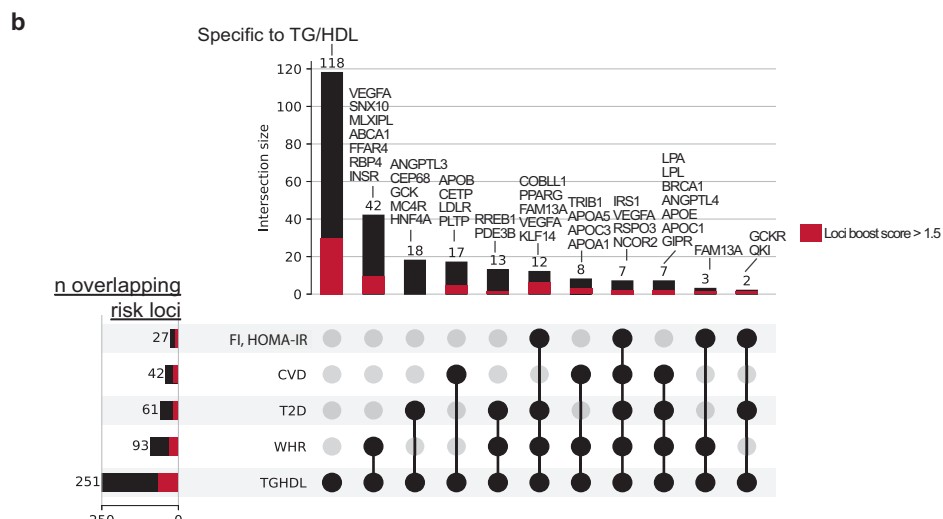

**Fig. 2 | Prioritizing "boosted" TG/HDL associations to identify insulin resistance loci. a** Cumulative distribution plot of 251 TG/HDL loci according to TG/HDL Boost score. Boost scores greater than 0 signify a stronger association with TG/HDL than TG or HDL alone. Boost scores less than 0 signify the opposite. Among the 251 TG/HDL loci, 109 (43%) have a boost score > 0 (dotted red line). Notable lipid regulation genes (TG:*ANGPTL3*, HDL:*CETP*) have highly negative boost scores, whereas known insulin resistance genes show positive boost scores. **b** Upset plot showing intersection of the 251 identified TG/HDL loci with previous GWAS for

surrogate insulin resistance markers and metabolic diseases: fasting insulin (FI), HOMA-IR, coronary artery disease (CVD), type 2 diabetes (T2D), waist-hip ratio (WHR). (Left) Total number of loci that overlap from each study with the 251 TG/HDL loci. (Top) Number of loci within each intersecting group of studies denoted by black circles with selected known insulin resistance loci labeled above. The proportion of each bar comprised of loci with top-quartile boost scores (>1.5) is highlighted in red. For example, TG/HDL, T2D, and FI share 2 loci, *GCKR* and *QKI*, in common.

common and distinct with our TG/HDL GWAS and top-quartile boosted loci (Fig. 2b). We scored the overlap of SNPs identified from studies of WHR (*n* = 694,649)[13], FI (*n* = 105,056)[14] or HOMA-IR (*n* = 37,037)[12], T2D (*n* = 180,834 cases vs. 1,159,055 controls)[41], and CVD (*n* = 181,522 cases vs. 1,165,690 controls)[47] with regard to their positional overlap with any of 251 TG/HDL loci identified in our study (Fig. 2b, Supplementary Data 4). WHR shared the largest number of overlapping associated loci with TG/HDL (*n* = 93 loci), followed by T2D (*n* = 61 loci), CVD (*n* = 42 loci), and loci associated with either FI or HOMA-IR (*n* = 27 loci). Of the 251 loci identified in our TG/HDL study, 118 had not previously been implicated as insulin resistance or metabolic disease risk loci (Fig. 2b). Considering the subset of the 62 top-quartile boosted loci, 29 had not previously been identified. These positional overlaps were largely concordant with formal genetic colocalization analysis (Supplementary Fig. 4, Supplementary Data 5).

## Putative causal gene identification and integrative genomics to infer direction-of-effect of "boosted" TG/HDL loci

For the top quartile of TG/HDL associated loci with positive boost scores (i.e., TG/HDL_BS > 1.5, *n* = 62 loci), we employed an integrative statistical fine-mapping approach instantiated in the "Sum of Single Effects" (SuSiE) model[48] ("Methods") to identify likely causal variants and genes. The SuSiE fine-mapping algorithm converged for 57 of the 62 loci, resulting in credible sets of SNPs with 95% posterior probability of containing a putative causal variant (Supplementary Data 6). Across the 57 mappable loci (i.e., algorithm converged to produce at least one credible set with one SNP), the defined credible sets contained a median of 66 variants (range 1–196); at 25 of these loci, at least one credible set consisted of only a single, high-confidence variant (posterior inclusion probability (PIP) > 95%).

Having defined the sets of potentially causal variants, we then used this information to nominate the most likely causal gene at each

locus applying commonly used practices[49]. Fifty fine-mapped loci contained at least one putative "causal" variant (defined as a variant with PIP > 0.1) (Table 1). These putative causal variants were cross-referenced with genomic and functional annotations (e.g., exons, promoters, expression quantitative trait associations) to assign a gene at each locus[49]. If the putative causal variant did not overlap with any known functional/genomic association, the nearest gene was assigned to be causal[49] (n = 27/50). This data-driven approach to identify candidate causal genes recovered several well-established insulin resistance genes at their respective risk loci, including *LPL*, *IRS1*, and *PPARG* (Table 1).

For each nominated candidate causal gene, we integrated diverse sources of gene-level evidence to identify the direction-of-effect on insulin resistance and evaluate tissue specificity (Fig. 3, Supplementary Data 7). To determine directionality with respect to gene function and TG/HDL levels in carriers, we utilized two complementary approaches leveraging rare and common genetic variation, respectively: (1) aggregation of rare, loss-of-function coding variants within each gene identified from exome sequencing of the UKBB participants (i.e., burden tests)[50] and (2) functional annotation of individual putative causal SNPs with sufficiently large MAF to permit individual association with predicted protein function or modification of gene expression (eQTLs). We further extended the analysis of eQTLs to all 50 putatively causal genes using established methods (Functional Summary-based Imputation (FUSION))[51] to correlate genetically predicted tissue-specific gene expression in metabolically relevant tissues (subcutaneous and visceral adipose, liver, and skeletal muscle) with TG/HDL in carriers.

This gene-level integrative genomics approach successfully recaptured the direction-of-effect and tissue specificity of several well-characterized insulin resistance effector genes, including *LPL*, *PPARG* and *ANGPTL4*, as well as a more recently characterized gene *COBLL1*. Gene-level burden tests showed that loss-of-function (LOF) protein-coding variation in *LPL* and *PPARG* is associated with an increase in TG/HDL (LPL: effect size = 0.04, $p = 3.1 \times 10^{-23}$; PPARG: effect size = 0.04, $p = 5 \times 10^{-6}$), consistent with the well-characterized roles of *LPL* and *PPARG* in insulin resistance[43,52]. At the *ANGPTL4* locus, fine-mapping identified causal SNPs rs116843064 (PIP = 1) and rs140744493 (PIP = 0.98) which encode missense variants in *ANGPTL4* (p.E40K and p.R336C respectively) that are computationally predicted by Ensembl Variant Effect Predictor (VEP)[53] to be deleterious. These SNPs were strongly associated with decreases in TG/HDL (rs116843064: effect size = −0.25, $p = 1.0 \times 10^{-189}$; rs140744493: effect size = −0.01, $p = 1.2 \times 10^{-5}$) in our data suggesting that LOF in *ANGPTL4* increases insulin sensitivity, a finding congruent with LOF mouse models of *ANGPTL4* resulting in reduced serum triglycerides and improved glucose homeostasis[28]. At the *COBLL1* locus, our genetically predicted gene expression analysis indicated that increased gene expression of *COBLL1* primarily in adipocytes (subcutaneous: t-statistic = −8.8, $p = 2 \times 10^{-16}$; visceral: t-statistic = −9.7, $p = 4.8 \times 10^{-20}$; liver: t-statistic = 3.0, $p = 0.03$; muscle: t-statistic = −3.5, $p = 0.007$) is associated with decreased TG/HDL which aligns with recent work showing that LOF in *COBLL1* results in insulin resistance through adipose-specific mechanisms[44].

### Evaluation of TG/HDL associated loci nominate *PLA2G12A*, *PLA2G6*, and *VGLL3* as novel insulin resistance genes

In an effort to identify novel insulin resistance genes, we focused our attention on loci that had not previously been associated with insulin resistance and metabolic diseases (Fig. 2b).

**PLA2G12A.** One such locus on chromosome 4, defined by lead SNP rs114816312 (effect size = 0.1, $p = 6 \times 10^{-25}$), was strongly boosted (TG/HDL$_{BS}$ = 3.7) and contained many genes (Fig. 4a). Fine mapping narrowed the credible set of SNPs to overlap with three genes: *PLA2G12A*, *MCUB* and *CASP6*. The lead variants with greatest

probability of being putatively causal both encoded missense variants in the *PLA2G12A* gene (rs114816312: p.D111N, PIP = 0.7; rs41278045: p.C131R, PIP = 1.0) strongly suggesting this as the candidate causal gene (Fig. 4a). *PLA2G12A* encodes a ubiquitously expressed (Supplementary Data 8), secreted member of the family of phospholipase A2 enzymes that hydrolyze phospholipids to release arachidonic acid[54]. Computational variant effect prediction with Ensembl Variant Effect Predictor (VEP)[53] predicted both putative causal missense variants to be deleterious to *PLA2G12A* protein function, and both variants were associated with increased TG/HDL levels suggesting that *PLA2G12A* promotes insulin sensitivity. As *PLA2G12A* was ubiquitously expressed but was not estimated to have significantly heritable gene expression, we therefore could not associate genetically predicted gene expression with TG/HDL values. Rather, we sought to independently corroborate the direction-of-effect with insulin sensitivity by investigating rare coding variation in the gene obtained from the UKBB exomes. Hypothesizing that LOF in *PLA2G12A* increases insulin resistance, we performed a series of genetic burden tests[55] aggregating LOF variants with decreasing stringency and computing association with TG/HDL. The most stringent filter for LOF variants (variants predicted deleterious by all five tools (see "Methods")) was associated with increased TG/HDL levels (effect size = 0.20, $p = 9 \times 10^{-21}$, Supplementary Data 9) supporting our hypothesis. As the stringency was relaxed ("Methods"), additional missense variants were included in the LOF group, increasing the significance of association between LOF in *PLA2G12A* and increased TG/HDL (Fig. 4b, Supplementary Data 9). Even after conditioning upon or removing the missense variants rs114816312 and rs41278045 described above, an independent exome-wide significant signal for association ($p < 2.5 \times 10^{-6}$)[56] between LOF in *PLA2G12A* and increased TG/HDL was identified (Fig. 4b, Supplementary Data 9).

**PLA2G6.** Another novel locus located on chromosome 22 marked by lead SNP rs200725415 (effect size = −0.02, $p = 9.3 \times 10^{-25}$) was also strongly boosted (TG/HDL$_{BS}$ = 4.0) (Fig. 4c). Fine-mapping of this locus produced only a single credible set containing variants with PIP > 0.1 which overlapped two genes *PLA2G6* and *MAFF*. However, all of the variants in the credible set with PIP > 0.1 were located within *PLA2G6*, nominating it as the candidate causal gene. *PLA2G6* encodes another member of the family of phospholipase A2 enzymes which hydrolyze phospholipids to release arachidonic acid but functions intracellularly[57].

In contrast to *PLA2G12A*, *PLA2G6* was estimated to have significantly heritable gene expression in skeletal muscle (heritability h² = 0.04, $p = 9.9 \times 10^{-4}$, see web resources[51]). Predicted skeletal muscle expression of *PLA2G6* was negatively associated with TG/HDL (t-statistic = −5, $p = 6.7 \times 10^{-7}$), suggesting increasing muscle expression of *PLA2G6* would potentially enhance insulin sensitivity. Concordantly, we found that genetically predicted muscle-specific *PLA2G6* expression was negatively associated with FI (t-statistic = −2.4, $p = 0.016$) in the MAGIC study (n = 105,056, Fig. 4d)[14]. To evaluate the potential consequence of muscle-specific *PLA2G6* expression on metabolic disease, we performed a similar association analysis with T2D (n = 180,834 cases vs. 1,159,055 controls)[41], CVD (n = 181,522 cases vs. 1,165,690 controls)[47], and NAFLD (n = 1483 cases vs. 17,781 controls)[40]. We found a significantly negative association of predicted gene expression with T2D only (t-statistic = −3.2, $p = 0.001$, Fig. 4d). These genetic analyses suggest that the heritable component of *PLA2G6* gene expression in muscle improves insulin sensitivity and reduces T2D risk.

To corroborate these genetic association analyses, we examined measured *PLA2G6* gene expression in biopsied skeletal muscle of 35 individuals undergoing insulin-sensitizing treatment with PPARG agonists, or thiazolidinediones (TZDs)[20]. Skeletal muscle *PLA2G6* expression was increased in paired samples after 3 months of TZD treatment relative to baseline (fold-change = 1.11, $p = 0.038$; Fig. 4e, Supplementary Data 10). We also identified a strong *PPARG* binding

**Table 1 | Nominated causal genes at top TG/HDL risk loci**

| Gene | Lead SNP | Chr:Pos | EA | OA | EAF | Beta (SE) | p-value | Boost score | $t_{sex\text{-}dimorphic}$ | $p\text{-}value_{sex\text{-}dimorphic}$ | Top Causal SNP | SNP-gene annotation |
|---|---|---|---|---|---|---|---|---|---|---|---|---|
| LPL | rs271 | 8:19813702 | A | G | 0.146 | −0.125 (0.003) | 3.58e-307 | 50.353 | −2.189 | 2.86e-02 | rs268 | Exonic (p.Asn318Ser) |
| ANGPTL4 | rs116843064 | 19:8429323 | A | G | 0.019 | −0.248 (0.007) | 1.03e-189 | 49.807 | −3.802 | 1.43e-04 | | Exonic (p.Glu40Lys); Regulatory variant |
| IRS1 | rs2943645 | 2:227099180 | T | C | 0.646 | 0.048 (0.002) | 2.48e-85 | 2.056 | 3.240 | 1.20e-03 | | Fine-mapped eQTL |
| APOB | rs533617 | 2:21233972 | C | T | 0.040 | −0.103 (0.005) | 5.14e-67 | 5.480 | −0.803 | 4.22e-01 | | Exonic (p.His1923Arg) |
| VEGFA | rs6905288 | 6:43758873 | A | G | 0.568 | 0.039 (0.002) | 8.54e-63 | 6.813 | 0.106 | 9.16e-01 | | Regulatory variant |
| KLF14 | 7:130438531_CTTTTTT_C | 7:130438531 | C | CTTTTTT | 0.516 | −0.037 (0.002) | 2.50e-55 | 12.202 | 7.959 | 1.73e-15 | | Nearest gene |
| COBLL1 | rs79953491 | 2:165555539 | G | A | 0.120 | −0.05 (0.003) | 2.78e-44 | 9.028 | 3.251 | 1.15e-03 | | Promoter |
| PLCB3 | rs71468663 | 11:64018104 | AC | A | 0.046 | 0.067 (0.005) | 1.23e-32 | 3.241 | −0.994 | 3.20e-01 | | Nearest gene |
| RSPO3 | rs6916318 | 6:127435106 | T | A | 0.531 | 0.028 (0.002) | 1.42e-32 | 5.971 | −5.274 | 1.34e-07 | rs57721086 | Exonic (5_prime_UTR_variant) |
| ARID1A | rs114165349 | 1:27021913 | C | G | 0.023 | 0.092 (0.007) | 3.13e-32 | 4.273 | −1.461 | 1.44e-01 | | Nearest gene |
| NTAN1 | rs11075253 | 16:15148646 | A | C | 0.297 | −0.03 (0.002) | 1.00e-30 | 3.781 | −0.415 | 6.78e-01 | | Promoter; Fine-mapped eQTL |
| PCCB | rs684773 | 3:135956305 | C | A | 0.767 | 0.031 (0.002) | 6.27e-30 | 2.487 | −0.741 | 4.59e-01 | | Regulatory variant |
| NEK4 | rs6800707 | 3:52516293 | G | C | 0.810 | 0.031 (0.003) | 5.37e-25 | 3.775 | −2.656 | 7.90e-03 | rs11235 | Exonic (3_prime_UTR_variant) |
| PLA2G12A | rs114816312 | 4:110638824 | T | C | 0.008 | 0.139 (0.012) | 6.43e-25 | 3.737 | 0.681 | 4.96e-01 | rs41278045 | Exonic (p.Cys131Arg) |
| PLA2G6 | rs200725415 | 22:38575498 | C | CT | 0.440 | −0.024 (0.002) | 9.26e-25 | 3.950 | −2.334 | 1.96e-02 | | Nearest gene |
| RBPJ | 4:26050450_AC_A | 4:26050450 | A | AC | 0.166 | 0.032 (0.003) | 1.01e-24 | 4.731 | −3.646 | 2.66e-04 | | Fine-mapped eQTL; Regulatory variant |
| PDE3A | rs11045171 | 12:20470199 | G | A | 0.198 | −0.029 (0.003) | 4.56e-23 | 4.616 | 1.173 | 2.41e-01 | rs4762753 | Fine-mapped eQTL |
| ACACB | rs149793040 | 12:109661672 | G | A | 0.002 | −0.262 (0.023) | 5.87e-23 | 1.906 | 0.336 | 7.37e-01 | | Exonic (p.Tyr1282Cys) |
| SNX10 | rs1534696 | 7:26397239 | A | C | 0.540 | −0.023 (0.002) | 3.93e-22 | 5.134 | 2.542 | 1.10e-02 | | Fine-mapped eQTL; Regulatory variant |
| TSC22D2 | rs62271373 | 3:150066540 | A | T | 0.060 | 0.046 (0.004) | 2.56e-20 | 5.194 | −0.016 | 9.88e-01 | | Nearest gene |
| AC022431.2 | rs60803019 | 5:558083415 | C | T | 0.065 | −0.043 (0.004) | 4.90e-20 | 3.422 | −0.853 | 3.94e-01 | rs455660 | Nearest gene |
| INSR | rs3890483 | 19:7220596 | T | G | 0.441 | 0.022 (0.002) | 5.41e-20 | 3.064 | −2.326 | 2.00e-02 | rs1799816 | Exonic (p.Val1012Met) |
| TTLL4 | rs148358468 | 2:219590348 | A | G | 0.050 | 0.047 (0.005) | 5.50e-18 | 3.488 | −0.348 | 7.28e-01 | rs116204487 | Promoter; Regulatory variant |
| TNFAIP8 | rs1045241 | 5:118729286 | T | C | 0.272 | −0.023 (0.002) | 5.53e-18 | 1.758 | 4.014 | 5.97e-05 | | Exonic (3_prime_UTR_variant) |
| RP11-1102P16.1 | rs13269725 | 8:72459889 | G | A | 0.079 | 0.037 (0.004) | 1.21e-17 | 2.402 | −3.241 | 1.19e-03 | | Exonic (5_prime_UTR_variant) |
| ADRB1 | rs2773469 | 10:115798895 | G | A | 0.735 | −0.022 (0.002) | 1.92e-17 | 2.904 | −0.204 | 8.38e-01 | | Nearest gene |
| PPARG | rs2067819 | 3:12359049 | A | G | 0.214 | −0.024 (0.002) | 2.04e-17 | 4.421 | −2.385 | 1.71e-02 | rs59447614 | Nearest gene |
| PEMT | rs11658944 | 17:17425279 | T | C | 0.057 | 0.042 (0.004) | 1.07e-16 | 3.618 | −3.067 | 2.16e-03 | | Nearest gene |
| USP3 | rs7175602 | 15:63367032 | C | T | 0.197 | 0.024 (0.003) | 5.84e-16 | 1.546 | 0.016 | 9.87e-01 | rs17184382 | Fine-mapped eQTL; Regulatory variant |
| SIPA1 | rs2306363 | 11:65405600 | T | G | 0.207 | −0.023 (0.003) | 7.64e-16 | 1.966 | −1.857 | 6.34e-02 | | Exonic (5_prime_UTR_variant) |
| KCNH2 | 7:150288766_GA_G | 7:150288766 | G | GA | 0.224 | 0.022 (0.002) | 2.39e-15 | 2.383 | −0.857 | 3.91e-01 | rs2968864 | Fine-mapped eQTL |
| CRTAC1 | rs563296 | 10:99772404 | A | G | 0.558 | 0.018 (0.002) | 2.71e-14 | 2.073 | −0.256 | 7.98e-01 | | Nearest gene |
| HNRNPK | rs296886 | 9:86592026 | G | A | 0.214 | −0.021 (0.002) | 7.79e-14 | 2.882 | 1.784 | 7.44e-02 | | Nearest gene |

**Table 1 (continued) | Nominated causal genes at top TG/HDL risk loci**

| Gene | Lead SNP | Chr:Pos | EA | OA | EAF | Beta (SE) | p-value | Boost score | $t_{sex\text{-}dimorphic}$ | $p\text{-value}_{sex\text{-}dimorphic}$ | Top Causal SNP | SNP-gene annotation |
|---|---|---|---|---|---|---|---|---|---|---|---|---|
| AHR | rs4410790 | 7:17284577 | C | T | 0.633 | 0.018 (0.002) | 3.71e-13 | 2.820 | -1.691 | 9.08e-02 | | Nearest gene |
| RP11-399J13.3 | rs678614 | 11:64799894 | C | A | 0.722 | -0.019 (0.002) | 7.56e-13 | 3.879 | -0.635 | 5.26e-01 | | Nearest gene |
| FARP2 | rs4675812 | 2:242395674 | A | G | 0.588 | -0.017 (0.002) | 1.53e-12 | 2.106 | 1.097 | 2.73e-01 | | Nearest gene |
| GADD45G | rs10797115 | 9:92191256 | T | C | 0.542 | 0.017 (0.002) | 1.59e-12 | 1.648 | -1.267 | 2.05e-01 | | Nearest gene |
| EYA2 | rs55966194 | 20:45599090 | G | C | 0.283 | -0.018 (0.002) | 3.59e-12 | 1.690 | 1.087 | 2.77e-01 | | Nearest gene |
| RNF168 | rs13094241 | 3:196190893 | G | T | 0.728 | -0.018 (0.002) | 9.51e-12 | 3.532 | 0.595 | 5.52e-01 | | Fine-mapped eQTL |
| VGLL3 | rs13066793 | 3:87037543 | G | A | 0.090 | -0.027 (0.004) | 1.98e-11 | 2.121 | 0.239 | 8.11e-01 | | Fine-mapped eQTL; Regulatory variant |
| CD248 | rs490972 | 11:66079786 | A | G | 0.469 | 0.015 (0.002) | 1.91e-10 | 2.400 | -0.621 | 5.35e-01 | rs1625595 | Fine-mapped eQTL |
| B3GNT4 | 12:122515547_ATTTTTC_A | 12:122515547 | A | ATTTTTC | 0.192 | 0.018 (0.003) | 1.06e-09 | 1.533 | -4.281 | 1.86e-05 | rs12827843 | Exonic (5_prime_UTR_variant) |
| ZIM2 | rs81102873 | 19:57488423 | T | C | 0.585 | 0.014 (0.002) | 1.43e-09 | 1.927 | -2.175 | 2.96e-02 | | Regulatory variant |
| AXL | rs4802113 | 19:41740895 | C | T | 0.457 | -0.014 (0.002) | 1.45e-09 | 1.884 | -1.125 | 2.60e-01 | | Nearest gene |
| RREB1 | rs35742417 | 6:7247344 | A | C | 0.184 | -0.018 (0.003) | 2.91e-09 | 1.986 | -0.358 | 7.20e-01 | | Exonic (p.Ser1554Tyr); Regulatory variant |
| UBE2K | rs145766974 | 4:39677849 | T | C | 0.569 | -0.013 (0.002) | 1.41e-08 | 2.346 | 2.409 | 1.60e-02 | rs34258469 | Nearest gene |
| QKI | rs4709746 | 6:164133001 | T | C | 0.134 | -0.019 (0.003) | 2.77e-08 | 2.129 | 0.675 | 5.00e-01 | rs7761309 | Regulatory variant |
| FGFR2 | rs878409 | 10:122999550 | A | G | 0.546 | -0.013 (0.002) | 3.39e-08 | 2.376 | 0.982 | 3.26e-01 | | Nearest gene |
| THRB | rs7640648 | 3:24313675 | G | A | 0.106 | 0.021 (0.003) | 3.63e-08 | 1.506 | 0.344 | 7.31e-01 | | Nearest gene |
| FADS2 | rs117186302 | 11:61456420 | C | T | 0.026 | -0.041 (0.007) | 4.36e-08 | 1.545 | -0.130 | 8.96e-01 | rs174566 | Nearest gene |

Fifty nominated genes at top quartile boosted TG/HDL risk loci. Each locus is defined by a lead SNP at Chr:Pos indicated, with positions in GRCH37. The effect/other alleles (EA/OA) are noted, and the effect allele frequency (EAF), Beta and standard error (SE), and GWAS p-value are in reference to the effect allele. The boost score is listed, which quantifies the difference in association of each locus between TG/HDL and TG or HDL alone, with boost score > 0 indicating increased significance in GWAS for TG/HDL compared to TG or HDL alone. The sex-dimorphic effects ($t_{sex\text{-}dimorphic}$) of each locus are shown, with a positive t-statistic indicating greater significance in female-specific GWAS for TG/HDL compared to male, and the sex-dimorphic p-value ($p\text{-value}_{sex\text{-}dimorphic}$) quantifying if the difference between sexes is significant. Top causal SNPs are listed if different from the locus lead SNP, and the annotation of the lead SNP/top causal SNP for gene nomination at each locus is listed.

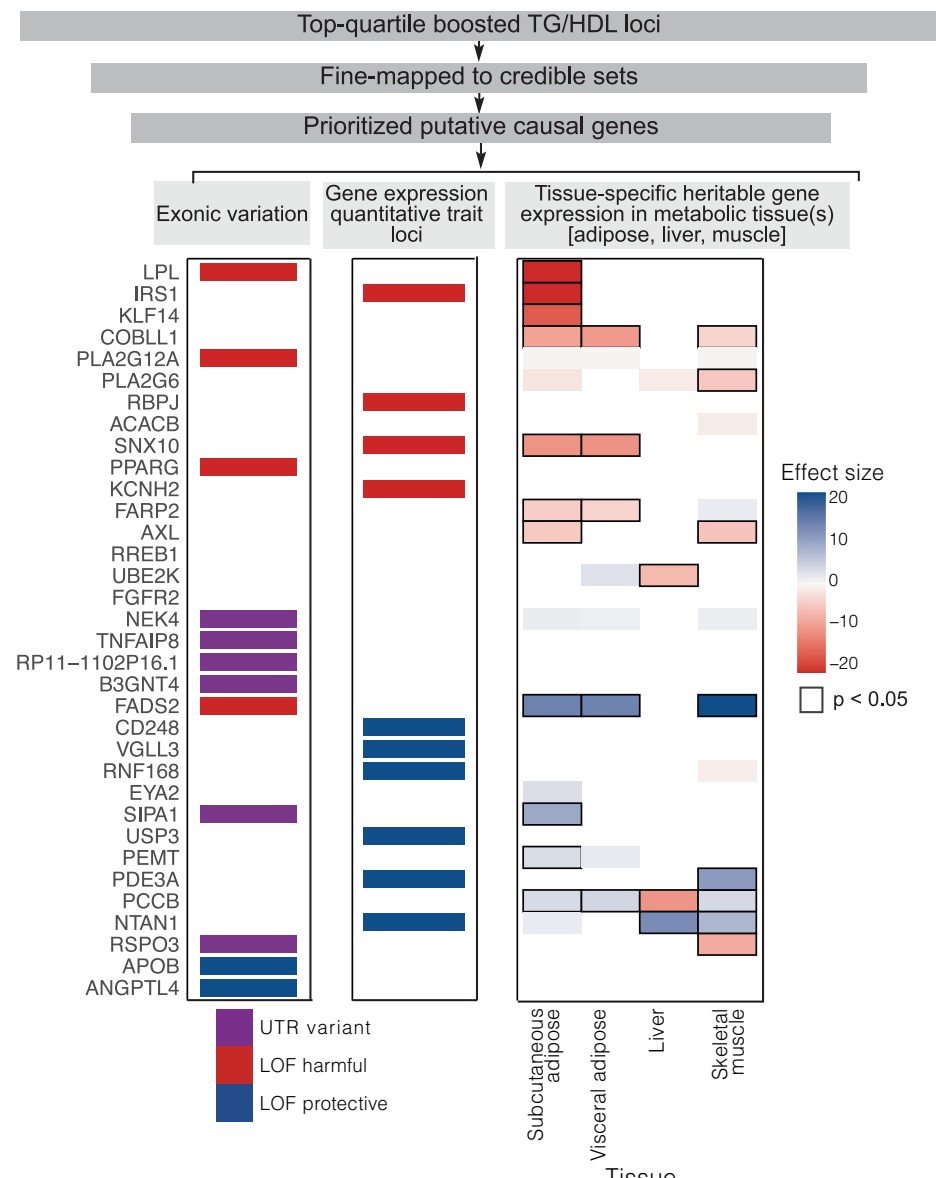

**Fig. 3 | Causal gene identification and integrative genomics to infer direction-of-effect of "boosted" TG/HDL loci.** Overview of the gene prioritization strategy for 251 TG/HDL associated loci. The top quartile of TG/HDL boosted loci ($n = 62$) was selected for fine mapping to obtain 95% credible sets of causal SNPs ($n = 57$). Of the loci with at least one causal variant identified from fine-mapping ($n = 50$), causal genes were assigned by incorporating genomic and functional annotations, including exonic variation and expression quantitative trait loci (eQTLs). Direction-of-effect on insulin resistance and tissue specificity of causal genes was assigned by examination of the following gene level evidence: (left) causal variants in exons were annotated as UTR (untranslated region; purple) or protein-coding with loss-of-function (LOF harmful: LOF increases TG/HDL (red); LOF protective: LOF decreases TG/HDL (blue)). (Middle) Causal variants that were eQTLs are assigned as LOF harmful or LOF protective if decreasing expression increases (red) or decreases TG/HDL (blue). (Right) Association of tissue-specific predicted gene expression with TG/HDL in the UK Biobank. All statistics were derived using a weighted linear regression model. Direction and magnitude of effect are represented by color, with positive and negative associations of gene expression with TG/HDL shown as blue and red, respectively, and darker shades demonstrating stronger effect sizes. Significant associations are outlined in black.

motif (JASPAR[58]: MA0066.1; chr22:38524661-38524680) in the intron of *PLA2G6* located 2231 bp upstream of the transcription start site (Fig. 4f). Taken together, these data suggest that *PLA2G6* may function through muscle to increase insulin sensitivity and decrease T2D risk.

**VGLL3.** A third novel TG/HDL locus identified on chromosome 3 was defined by lead SNP rs13066793 (effect size = −0.03, $p = 2 \times 10^{-11}$, Fig. 4g), which was located within an intron of the gene *VGLL3* and annotated as an eQTL for *VGLL3* expression in muscle (effect size = −0.25, $p = 8 \times 10^{-7}$) and nominally in subcutaneous adipose

(effect size = −0.1, $p = 0.02$). These data suggest that increased expression of *VGLL3*, which encodes a transcriptional co-activator for TEAD family transcription factors[59], increases insulin resistance. We note that despite the highly significant muscle eQTL, *VGLL3* was lowly expressed in human skeletal muscle (median transcripts per million (TPM): muscle = 0.9 TPM), whereas it was highly expressed in human adipose tissues (subcutaneous adipose: 24.8 TPM, visceral adipose: 17.4 TPM, GTEx[60]: Supplementary Data 8). Thus, we attempted to corroborate adipose *VGLL3* expression with insulin sensitivity in transcriptomically profiled adipose and skeletal muscle tissue biopsies obtained from 31 individuals undergoing hyperinsulinemic-

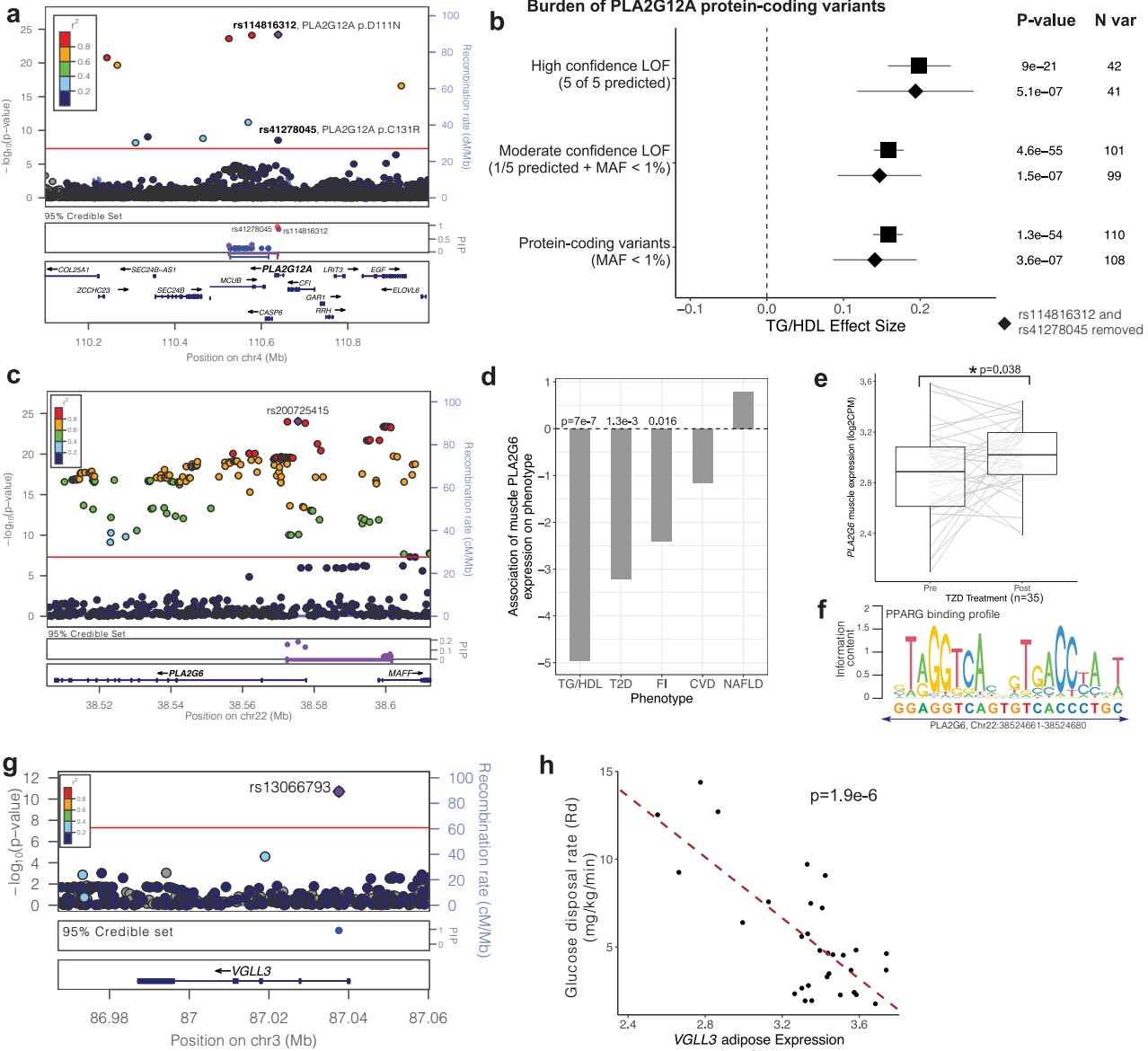

**Fig. 4 | Evaluation of TG/HDL-associated loci nominates *PLA2G12A*, *PLA2G6*, and *VGLL3* as novel insulin resistance genes. a** LocusZoom plot of the *PLA2G12A* TG/HDL locus. The purple diamond represents the lead SNP, and color represents the linkage disequilibrium between each variant with the lead SNP. The *PLA2G12A* locus contains three 95% credible sets (depicted as red, purple, blue), with the top two causal SNPs rs114816312 (PIP = 0.7) and rs41278045 (PIP = 1) encoding *PLA2G12A* missense variants. **b** *PLA2G12A* loss-of-function (LOF) burden tests in the UK Biobank exome sequenced individuals with increasing confidence (stringency) to define LOF variants. High confidence LOF variants were called deleterious by 5 out of 5 bioinformatics prediction tools (see "Methods"), Moderate confidence LOF variants were those called deleterious by at least 1 bioinformatic prediction tool and had minor allele frequency (MAF) < 1%. Standard burden test results are indicated by squares, and diamonds represent sensitivity analyses removing the top causal variants rs41278045 and rs114816312. 95% confidence intervals (±1.96*standard error) are shown as whiskers. *P*-values (linear ridge regression models) and

number of variants for each analysis are listed. **c** LocusZoom plot and 95% credible set for the *PLA2G6* TG/HDL locus. **d** Associations of predicted muscle *PLA2G6* expression on TG/HDL, type 2 diabetes (T2D), fasting insulin (FI), coronary artery disease (CVD), and non-alcoholic fatty liver disease (NAFLD) (Bayesian sparse linear mixed model). **e** Upregulation of skeletal muscle *PLA2G6* upon TZD treatment in 35 biopsied clinical study participants (fold-change = 1.11, *p = 0.038; linear mixed model). For each boxplot, the middle line in the box indicates the median, the box boundaries represent the 25th and 75th percentiles, and the whiskers are the 5th and 95th percentiles. **f** *PPARG* consensus motif overlaid over a high-scoring matching sequence 2231 bp 5' of the *PLA2G6* transcription start site. **g** LocusZoom plot and 95% credible set for the *VGLL3* TG/HDL locus. **h** Association of adipose tissue *VGLL3* expression with glucose disposal rate (Rd) measured by hyperinsulinemic-euglycemic clamp among 31 biopsied participants (b = −8.7, p = 1.9e-6; linear model).

euglycemic glycemic clamps[20]. We identified a significantly negative correlation between glucose disposal rate (Rd) and adipose tissue *VGLL3* expression (estimate = −8.7, *p* = 1.9 × 10⁻⁶, Fig. 4h, Supplementary Data 11), but no significant correlation with skeletal muscle *VGLL3* expression (estimate = 1.2, *p* = 0.56). We additionally adjusted gene expression for subject-specific covariates and found no effect for age

or sex but a strong effect for BMI (estimate = −0.24, *p* = 3.2 × 10⁻⁴). Formal mediation analysis comparing regression models of Rd against adipose *VGLL3* expression with and without BMI revealed approximately 50% of the effect of *VGLL3* adipose expression on glucose disposal is mediated by BMI. Taken together, these data suggest that *VGLL3* increases insulin resistance by altering adiposity.

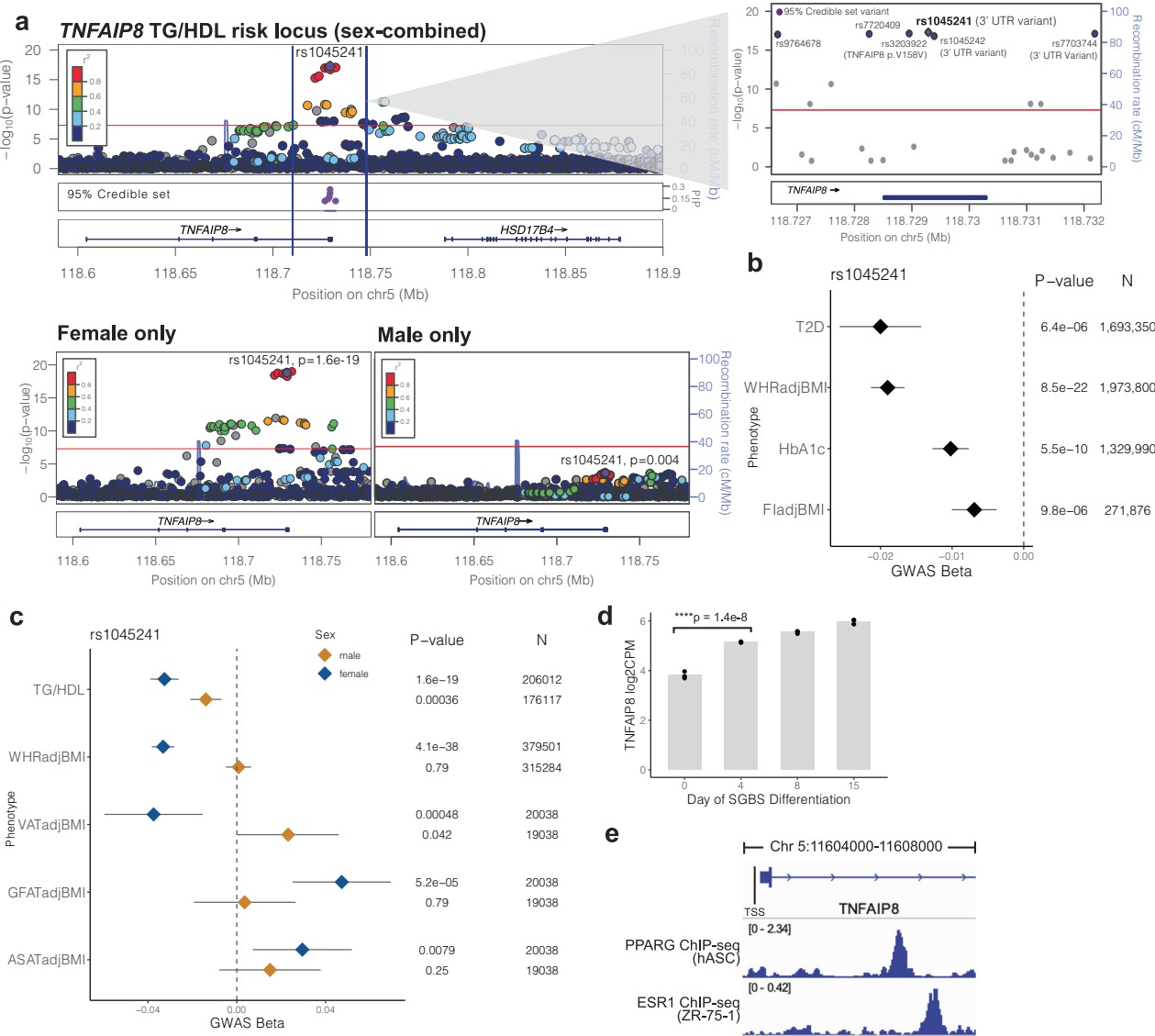

**Fig. 5 | *TNFAIP8* increases insulin resistance through the alteration of adipose tissue depots in a female-specific manner.** WHR waist-hip ratio, ASAT abdominal subcutaneous adipose, VAT visceral adipose, GFAT gluteofemoral adipose, T2D type 2 diabetes, FI fasting insulin, TG triglycerides, HDL HDL cholesterol, GGT gamma glutamyl transferase, HbA1c glycated hemoglobin, ApoA apolipoprotein A. **a** (top) LocusZoom plot of the TG/HDL associated locus containing *TNFAIP8* from the sex-combined GWAS. The lead SNP (rs1045241, shown as purple diamond) and GWAS *p*-value (whole-genome regression model) are labeled, and color represents the LD ($r^2$) between each variant with the lead SNP. (Inset panel) Zoom in of the 95% credible set region consisting of 6 SNPs (purple). (Bottom) LocusZoom plots of the *TNFAIP8* TG/HDL locus from the sex-stratified GWAS. The association is only significant in females. **b** Association of *TNFAIP8* lead SNP rs1045241 in meta-analyzed metabolic disease/trait association studies. T2D, WHR, FI and HbA1c are significantly decreased by the TG/HDL lowering effect allele. Effect estimates (beta)

and 95% confidence intervals are plotted as diamonds and whiskers, along with association *p*-values (fixed-effect meta-analysis) and sample sizes. **c** Sex-stratified associations of rs1045241 with metabolic phenotypes. Effect estimates (beta) and 95% confidence intervals are plotted as diamonds and whiskers, and shown for female (blue) and male (orange) stratified GWAS, along with association *p*-values (linear mixed model) and sample sizes. The TG/HDL lowering effect allele decreases VAT and increases GFAT and ASAT in a female-specific manner. **d** Upregulation of *TNFAIP8* expression in human pre-adipocytes during in vitro differentiation. Linear mixed model was used to obtain the *p*-value; *n* = 3 biological replicates per time point. **e** Gene model of *TNFAIP8* highlighting the transcriptional start site (TSS) and first intron (top). (Second track) *PPARG* ChIP-seq profiling in human adipocytes (human adipose stem cells, hASC). (Bottom) *ESR1* ChIP-seq profiling in human breast cell line ZR-75-1.

## *TNFAIP8* increases insulin resistance through the alteration of adipose tissue depots in a female-specific manner

Among the TG/HDL "boosted" association signals, a locus on chromosome 5 marked by lead SNP rs1045241 stood out as demonstrating strong evidence for sexual dimorphism, indicating a stronger effect in females ($t_{sex}$ = 4.0, $p_{sex}$ = 6.0 × 10⁻⁵; Supplementary Data 1). Sex-stratified analysis in the female (*n* = 206,012) and male (*n* = 176,117) UKBB participants showed that the association with TG/HDL was highly female-specific with little evidence of association in the

subgroup of male participants (effect size$_{female}$ = −0.03, $p_{female}$ = 1.6 × 10⁻¹⁹; effect size$_{male}$ = −0.01, $p_{male}$ = 3.5 × 10⁻⁴; Fig. 5a, Supplementary Data 1). In fact, despite half the sample size, the association was stronger in the subgroup of female participants than in the sex-combined discovery cohort (*n* = 382,129). Fine-mapping at the locus produced a single credible set with six putatively causal variants, all of which were located within the gene *TNFAIP8* (Fig. 5a, inset panel). The lead variant rs1045241 (chr5:118729286:C:T) had the highest probability of being putatively causal (PIP = 0.24) and was located in the 3'

UTR of *TNFAIP8*. Overall, four of the six variants were located within this last exon of *TNFAIP8*—one coding and three in the 3' UTR (Supplementary Data 6, Fig. 5a) strongly nominating it as the candidate causal gene. *TNFAIP8* encodes the tumor necrosis factor accessory protein 8 which has been found to play an anti-apoptotic role for TNFalpha-induced tumor apoptosis[61], but has no known role in insulin resistance or metabolic disease.

In a serum proteomics study of 35,559 individuals[62], the effect allele (chr5:118729286:C:T) of the lead putatively causal variant rs1045241, which was associated with decreased TG/HDL also was significantly associated with decreased plasma levels of *TNFAIP8* (effect size = −0.05, $p = 8.7 \times 10^{-10}$), suggesting *TNFAIP8* promotes insulin resistance. To corroborate this direction-of-effect, we examined other metabolic disease/trait associations for rs1045241 (meta-analyzed within the Common Metabolic Diseases Knowledge Portal (see Web Resources)) and found that the effect allele (chr5:118729286:C:T) was also associated with decreased waist-to-hip ratio (WHRadjBMI effect size = −0.02, $p = 8.5 \times 10^{-22}$, n = 1,973,800), decreased FIadjBMI (effect size = −0.01, $p = 9.8 \times 10^{-6}$, n = 271,876), decreased glycated hemoglobin (HbA1c; effect size = −0.01, $p = 2.7 \times 10^{-9}$, n = 1,329,990) and decreased risk for T2D (odds ratio = 0.98, $p = 6.4 \times 10^{-6}$, n = 1,693,350) (Fig. 5b). These complementary associations support an insulin-sensitizing effect from decreasing serum TNFAIP8 levels.

To further corroborate these findings, we directly examined TNFAIP8 levels in 54,219 serum samples from the UK Biobank generated using the Olink Explore 3072, a platform for high throughput proteomics that quantified 2945 proteins[63]. We found that plasma TNFAIP8 levels correlated with increased TG/HDL, HbA1c, and WHR levels, with a notably stronger association in females than in males ($p_{TG/HDL:female} < 2 \times 10^{-16}$, $p_{TG/HDL:male} = 3.23 \times 10^{-4}$, Supplementary Data 12).

We further investigated the WHR association where the UK Biobank-GIANT consortium[13] had made available sex-stratified summary statistics. rs1045241 had a much stronger association with WHRadjBMI in females ($p_{female} = 4.1 \times 10^{-38}$) and no evidence of association in males ($p_{male} = 0.8$) (Fig. 5c). As WHRadjBMI is a surrogate measure for enhanced visceral adiposity[64], we examined the association of rs1045241 with MRI quantified visceral adipose tissue (VATadjBMI), gluteofemoral adipose tissue (GFATadjBMI), and abdominal subcutaneous adipose tissue (ASATadjBMI) from a prior study of 19,038 male and 20,038 female participants in the UKBB[65]. We found that the effect allele (chr5:118729286:C:T) that lowered both TG/HDL and WHR with a female-biased effect concordantly decreased VATadjBMI and increased GFATadjBMI in females (effect size$_{VAT}$ = −0.4, $p_{VAT} = 4.8 \times 10^{-4}$; effect size$_{GFAT}$ = 0.05, $p_{GFAT} = 5.2 \times 10^{-5}$; n = 20,038) but not in men (Fig. 5c, Supplementary Data 13). ASATadjBMI was also increased in women (effect size$_{ASAT}$ = 0.03, $p_{ASAT} = 7.9 \times 10^{-3}$) but not in men. The sexual dimorphism in these associations was so strong that the association signal was lost in a sex-combined analysis of rs1045241 with VATadjBMI ($p = 0.2$, n = 39,076) despite doubling the number of samples. These data suggest that *TNFAIP8* may operate in adipose tissue in a sex-specific manner to promote insulin resistance.

To corroborate these findings at a cellular and molecular level, we analyzed *TNFAIP8* expression levels during the course of human adipocyte differentiation and found it to be significantly upregulated (fold-change = 2.5, $p = 1.4 \times 10^{-8}$) during the conversion of pre-adipocytes into mature adipocytes in vitro[66] (Fig. 5d). We analyzed data of chromatin immunoprecipitation and DNA sequencing (ChIP-seq) of *PPARG*[67], a master regulator of adipocyte differentiation[68], and found a strong peak of binding (chr5:118606764-118607039) 2377 base pairs downstream of the *TNFAIP8* transcriptional start site in the first intron (Fig. 5e). Given the female-specific association of *TNFAIP8*, we next examined publicly available estrogen receptor (*ESR1*) ChIP-seq data at the *TNFAIP8* locus and identified a strong binding peak of binding (chr5:118607354-118607660) in human breast cell line ZR-75-

1[69] approximately 600 base pairs downstream of the *PPARG* binding site. This points to a potential molecular mechanism for the female-specific signals of association observed between *TNFAIP8* and metabolic traits.

## Discussion

In this study, we conducted a GWAS for TG/HDL, a surrogate marker of insulin resistance, identifying 251 associated loci, of which 62 associated more strongly with TG/HDL compared to TG or HDL alone, and 118 which had not been previously reported in prior insulin resistance GWAS studies[12–14,41,47]. At 50 of these loci, we identified causal variants and genes performing integrative genomic analysis to propose directions-of-effect and tissue of action. For selected genes, we independently corroborate these genetic findings by interrogating gene expression in biopsied skeletal muscle and adipose tissue in individuals with directly measured insulin resistance by glycemic clamp. We highlight two phospholipase encoding genes, *PLA2G12A* and *PLA2G6*, which enhance insulin sensitivity, a transcriptional co-activator *VGLL3* which decreases insulin sensitivity, and a TNFα accessory protein *TNFAIP8* which decreases insulin sensitivity specifically in females.

The re-identification of loci known to encode insulin signaling components (e.g., *INSR*, *IRS1*), as well as those identified by previous surrogate biomarker studies, confirms the fidelity of TG/HDL as a surrogate marker of insulin resistance, providing confidence that the 29 novel loci in the top-quartile of boost scores identified could be true mediators of insulin resistance. Compared to prior surrogate marker studies, the new signals identified in our study could arise for two reasons: (1) sample size and (2) physiological/biological heterogeneity. Regarding sample size, our study (n = 382,129) is larger than other serum derived biomarkers (FI: n = 105,056[14]), but is exceeded in sample by anthropometric biomarker studies (WHR: n = 694,649[13]). If sample size related statistical power were a major determinant of locus identification, we would expect that association strength for known causal insulin resistance genes (e.g., *PPARG*, *IRS1*) would be proportional to study size, but in fact found no such relationship for *PPARG* ($p_{PPARG-FI} = 1.5 \times 10^{-21}$, $p_{PPARG-TG/HDL} = 2 \times 10^{-17}$, $p_{PPARG-WHR} = 3.8 \times 10^{-25}$) or *IRS1* ($p_{IRS1-FI} = 8.5 \times 10^{-39}$, $p_{IRS1-TG/HDL} = 2.5 \times 10^{-85}$, $p_{IRS1-WHR} = 1.6 \times 10^{-9}$; Supplementary Data 1 and 4). Regarding biological/physiological heterogeneity, insulin is known to signal in a tissue-specific manner—e.g., the level of insulin required physiologically to suppress ketogenesis or lipolysis is orders of magnitude less than required to induce glucose uptake[70]. Thus, each surrogate biomarker may capture overlapping but distinct aspects of insulin resistance physiology. Surrogate biomarker GWAS demonstrating differing patterns of tissue enrichments of associated loci support this idea. The loci associated with TG/HDL in our study are most highly enriched in adipose tissue, followed by liver and skeletal muscle (Supplementary Fig. 3), whereas for FI (skeletal muscle enrichment = adipose > liver) and for FG (liver enrichment > skeletal muscle > adipose)[11]. Taken together, these observations suggest surrogate biomarker studies of insulin resistance capture complementary aspects of physiology, and distinct loci from these studies could point to new mechanisms of molecular pathogenesis.

Among the loci uniquely associated with TG/HDL, we prioritized two phospholipase A2 enzymes, *PLA2G12A* and *PLA2G6*, that function in the release of arachidonic acid, a precursor of the bioactive lipid molecules eicosanoids[71]. Eicosanoids are known to be involved in inflammation and insulin sensitivity[72], suggesting dysregulation of PLA2s may increase insulin resistance and related metabolic disorders via disruption of eicosanoid signaling. Through examination of rare protein-coding variation from exome sequencing, we find that loss-of-function in *PLA2G12A* increases insulin resistance as measured by TG/HDL. *PLA2G12A* is secreted to the blood, ubiquitously expressed across tissues, and does not have any tissue-specific associations, indicating that it may function across several tissues to affect whole-body insulin sensitization. On the other hand, our analyses propose a

tissue-specific role for *PLA2G6* in muscle under the regulation of *PPARG* to increase insulin sensitivity and decrease T2D risk (Fig. 4d–f). The metabolic role of *PLA2G6* in pancreatic islets has previously been examined in vitro and in murine models[73], but to our knowledge, this is the first study implicating it in human insulin resistance.

Another uniquely associated TG/HDL locus is *VGLL3*, a transcriptional cofactor[74] whose adipose expression is inversely correlated with glucose disposal rate in humans suggesting *VGLL3* promotes insulin resistance (Fig. 4h). We also found that adipose *VGLL3* expression is correlated with human BMI and this accounts for about half of the effect on clamp-based glucose disposal. In contrast to our findings in humans, previous investigation in mice has shown that *VGLL3* expression is negatively correlated with murine adiposity[59] and that *VGLL3* can suppress murine adipocyte differentiation in vitro. Further investigation, including gain and loss-of-function studies in human adipocyte models, would shed light on this interspecies discordance.

Through sex-stratified GWAS, we prioritized *TNFAIP8* as a female-specific insulin resistance gene that promotes visceral adipose tissue accumulation at the expense of subcutaneous depots (GFAT and ASAT, Fig. 5c). Prior investigators have identified TNFAIP8 as an anti-apoptotic protein induced in immune cells by TNFα[75] capable of binding free fatty acids. Its reported biological function has been as an inducer of autophagy and steatosis in an oncogenic context, potentially through interaction with ATG family proteins and inhibition of the AKT/mTOR pathway[61]. Our study reveals a sex-dimorphic functional role for *TNFAIP8* in adipocytes potentially through the control of PPARG and estrogen (Fig. 5d, e) that increases insulin resistance and T2D risk, specifically in women. Furthermore, we find that serum levels of *TNFAIP8* are potentially causal for these effects. These findings warrant further investigation into the downstream mechanisms of *TNFAIP8* in adipose insulin resistance and evaluation of its potential as an intervenable biomarker for insulin resistance-related diseases in women.

A recent publication by Oliveri et al.[76] also examined the genetic association of TG/HDL in the UK Biobank, providing an opportunity for direct comparison. Despite including different numbers of samples ($n = 402,398$ (Oliveri) v.s. $n = 382,129$ in our study) due to some methodological differences in sample selection (e.g., Oliveri et al. included white European participants only) and QC (e.g., we performed sample level filtering as per the GLGC standard protocols[21,22]; Supplementary Fig. 1B), we found a high degree of concordance in the identified signals. In total, 241 of the 251 independent loci we identified (Supplementary Data 1) were matched directly or via proxy SNPs to Oliveri with a strong correlation in effect sizes ($r = 0.986$, $p < 0.001$, Pearson correlation) and *p*-values ($r = 0.985$, $p < 0.001$, Pearson correlation). Of the remaining variants not found in the Oliveri study, 9 were on the X chromosome, which was specifically excluded from analysis in Oliveri et al., and one rare variant (rs138611280; MAF = 0.001) could not be matched. In addition to including X chromosome variants, our study importantly differs due to the inclusion of indel variants and low-frequency variants (MAF $\geq$ 0.001 vs. Oliveri MAF $\geq$ 0.01) which impact gene identification. For example, at the *PLA2G6* locus (Fig. 4c), the lead variant identified in our study is a deletion (rs200725415, 22:38575498:CT:C) and has the highest PIP for being a causal variant in fine mapping analysis (Supplementary Data 6). While this locus is also identified in Oliveri et al. via the proxy SNP (rs2267373, 22:38600542:C:T), it does not meet the threshold (PIP > 0.1) for a causal variant, and the more likely causal variant identified by our study is filtered out in Oliveri et al. because it is a deletion. Similarly, at the *PLA2G12A* locus, the higher MAF threshold employed by Oliveri filters out causal coding SNPs (rs114816312, 4:110638824:C:T; Fig. 4a).

Our study has several important limitations that should be considered. First, we used the TG/HDL ratio as a surrogate marker of insulin resistance, which may not reflect the true causal relationship between genetic variants and insulin resistance as is the case with other surrogate marker GWAS[12–14]. We believe this trade-off is reasonable given the vast increase in sample size possible through the use of TG/HDL and its strong phenotypic and genetic correlations with directly measured insulin resistance and metabolic diseases (Supplementary Fig. 1A and Fig. 1). These genetic correlations are observed even though the UK Biobank TG/HDL ratio was computed from non-fasting measurements, which could confound the use of TG/HDL as a surrogate insulin resistance marker. Moreover, the TG/HDL ratio is influenced by the individual levels of TG and HDL, which share loci with the TG/HDL ratio and could result in true locus associations but false positives for insulin resistance. Therefore, we prioritized "boosted" genes that showed a stronger association with the TG/HDL ratio than with TG or HDL alone, but we cannot exclude the possibility that some of these genes are also involved in lipid metabolism (e.g., *APOB*, TG/HDL$_{BS}$ = 5.5). Second, we inferred the tissue specificity of the genes based on their heritable expression patterns in metabolic tissues, which have some limitations. For example, gene expression can be correlated across different tissues, making it difficult to identify the primary tissue of action for each gene[77]. Also, due to linkage disequilibrium, the genetic determinants of gene expression (eQTLs) may not be specific to the gene of interest but may tag other nearby or distant genes that have functional effects on insulin resistance. Furthermore, the heritability of gene expression is sensitive to sample size and may be low or nonsignificant for many genes in relevant metabolic tissues. In our analysis, we find nonsignificant heritability estimates for over half (52.5%) of our 50 genes of interest in relevant metabolic tissues, not a surprising proportion given less than half of all genes have significantly heritable gene expression in blood at existing sample sizes[78]. Therefore, while we were able to corroborate tissue specificity for several genes highlighted in this study (*PLA2G6*, *VGLL3*, *TNFAIP8*), our tissue specificity analysis cannot be considered comprehensive for the nominated genes. Further study of these genes is needed to elucidate the causal mechanisms and target tissues underlying the modulation of insulin resistance.

In conclusion, our study reports on GWAS of TG/HDL ratio at population scale expanding our knowledge of the genetic determinants of insulin resistance to include dozens of new candidate loci/genes not previously implicated. At four loci *PLA2G12A*, *PLA2G6*, *VGLL3, and TNFAIP8*, we validate direction-of-effect, potential regulatory mechanisms, and sex-specific effects. These findings provide specific, testable hypotheses for future investigation to credential these genes as potential new therapeutic targets and/or biomarkers for insulin resistance.

## Methods

Our research complies with all relevant ethical regulations. The UK Biobank study was approved by the Research Ethics Committee, and informed consent was obtained from all participants. Analysis of UK Biobank data was conducted under application numbers 51436 and 26041. The experimental protocol for tissue biopsies of subcutaneous adipose tissue and skeletal muscle was approved by the Institutional Review Board for Human Subjects of the University of California, San Diego, and informed consent was obtained from all participants[20].

### Phenotypic correlations of TG/HDL with insulin resistance biomarkers in a clinical cohort

To assess the relationship between TG/HDL and other surrogate insulin resistance phenotypes, we examined a clinical cohort of 45 human participants who underwent hyperinsulinemic-euglycemic clamp and additionally had other metabolic phenotypes measured[20]. We included participants across all insulin sensitivity classes and only included biological samples from before TZD treatment in the fasting state, pre-glycemic clamp. Pairwise correlations of TG/HDL with glucose disposal rate measured by hyperinsulinemic-euglycemic clamp,

fasting insulin, and HOMA-IR were assessed using Spearman rank correlation.

### Genome-wide association analyses

**UK Biobank data.** Analysis of UK Biobank data was conducted under application numbers 51436 and 41189. The UK Biobank is a prospective cohort study with genotypic and phenotypic data that enrolled approximately 500,000 individuals aged 40–69 from across the United Kingdom[79]. To remove potential outliers and individuals with poor data quality, we applied several sample-level filters to the UK Biobank participants following the published standards[21,22]. Specifically, we retained only those individuals with a sample call rate greater than 95% and with heterozygosity values less than the median + 3x the interquartile range (IQR) and excluded those with reported gender-genetic sex mismatch (Supplementary Fig. 1B). After filtering, 382,129 UK Biobank participants with TG and HDL measurements available remained of which 206,012 were female and 176,117 were male.

For quality control of SNPs from the UK Biobank genotypes, we included only variants with SNP call rate > 98%, MAF > 0.001, HWE $p > 5 \times 10^{-8}$, and for imputed SNPs included those with INFO scores >0.4. For imputation, the Haplotype Reference Consortium (HRC) and UK10K / 1000 Genomes reference panels were used as described in Bycroft et al.[79] We excluded duplicated SNPs and removed any monomorphic markers. After filtering, 489,897 genotype array SNPs and 12,024,213 imputed SNPs remained.

**Association testing using REGENIE.** REGENIE was run following the published recommendations for UK Biobank analysis (https://rgcgithub.github.io/regenie/recommendations/) to conduct genome-wide association analyses for TG/HDL, TG, and HDL using the 382,129 participants and 12,024,213 SNPs which passed QC. In brief, TG and TG/HDL were natural log-transformed, and all phenotypes (TG, HDL, and TG/HDL) were inverse rank normalized using the REGENIE --apply-rint parameter. In the statistical models, the covariates included were age, age[2], sex, and the first 20 principal components of ancestry to correct for population stratification. As recommended, Step 1 of REGENIE was first run using the 489,897 genotype array SNPs which passed QC to fit the whole-genome regression model to the phenotypes and produce genomic predictions, then Step 2 was run to perform association testing of the 12,024,213 imputed SNPs. We next estimated the genomic inflation for the TG/HDL GWAS (λ = 1.35) and subsequently performed correction for inflation using the genomic control method[80]. Lead SNPs and genomic risk loci from the conducted GWAS were then defined using the FUMA platform using the default parameters[24,80].

**Replication GWAS for TG/HDL in Mass General Biobank (MGBB).** GWAS for TG/HDL was conducted in the MGBB (*n* = 37,545) using consistent methodology as conducted in the UK Biobank. The MGBB is an enterprise-wide biobank that has enrolled patients aged 20–99 from across Mass General Brigham, a healthcare system located primarily in the Boston area of the United States[36]. Lead SNPs of each of the 251 TG/HDL genomic risk loci were examined in the MGBB, and if the lead SNP was not present in the MGBB, the next most significant SNP within the locus clump was used as a proxy SNP for replication (Supplementary Data 1). Statistical power for replication in the MGBB was calculated using the power calculator for genetic association studies (R genpwr v1.0.4)[81].

**Sex stratified analysis.** Sex-stratified GWAS for TG/HDL were conducted following the same methods above except with sex removed as a covariate and the --sex-specific female/male REGENIE parameter added. To quantify sex differences in genetic effects of each TG/HDL risk locus defined in the sex-combined GWAS, we calculated the t-statistic and *p*-value (by two-tailed Student's *t*-test) for the lead SNP of each locus using the sex-stratified GWAS beta estimates and standard errors as previously described[82].

### Tissue enrichment of TG/HDL genomic risk loci

We used GREGOR (v.1.4.0)[31] to determine the enrichment of the identified TG/HDL genomic risk loci that overlapped tissue-specific stretch enhancers, using the lead SNP at each locus as input. Genomic regions of stretch enhancers in each tissue were defined as in Chen et al.[11]. The GREGOR default parameters were used to run enrichment analyses ($r^2$ threshold = 0.8, LD window size = 1 Mb, and minimum neighbor number = 500).

### Examining the genetic overlap between TG/HDL and insulin resistance phenotypes

We assessed the genetic correlation between TG/HDL and insulin resistance phenotypes using cross-trait LD Score regression with the default settings as instantiated in LD SCore (LDSC v1.0.1)[37,83]. All association statistics were harmonized prior to computing genetic correlations, ensuring consistent genome build and effect alleles across studies.

We further examined the positional overlap between genomic risk loci associated with TG/HDL with other previously identified insulin resistance-related genetic risk loci. Overlap of genomic risk loci was defined if a lead SNP defining an insulin resistance-related locus fell within the locus boundaries as defined by FUMA of a TG/HDL genomic risk locus identified in our GWAS (Supplementary Data 4).

We further assessed for the presence of genetic overlapping signals between genomic risk loci associated with TG/HDL and associations for CVD[42], T2D[41], WHR[13], and FI[14] using COLOC[84]. Suggestive overlapping signal at a locus between TG/HDL and insulin resistance-related phenotype was defined with a threshold of posterior probability Bayesian factor H3 (PP.H3.abf) + posterior probability Bayesian factor H4 (PP.H4.abf) ≥ 0.99.

### Computing TG/HDL boost score

To quantify the difference in the association between TG/HDL compared to TG or HDL alone, a SNP-wise "boost score" (TG/HDL$_{BS}$) was computed from GWAS performed in the UK Biobank for TG, HDL and TG/HDL. For each lead SNP in the TG/HDL study, the more significant association *p*-value (-log10(p)) between TG or HDL was subtracted from the -log10(*p*-value) of the TG/HDL association. TG/HDL$_{BS}$ = $-\log10(p_{TG/HDL}) - \max\{-\log10(p_{TG}), -\log10(p_{HDL})\}$.

### Fine-mapping and nomination of top candidate causal gene at top quartile boosted loci

To determine causal SNPs at each TG/HDL genomic risk locus, statistical fine-mapping was performed using the "Sum of Single Effects" (SuSie) model[48] using default parameters. UK Biobank genotypes were down-sampled to 10,000 individuals to be used as an LD reference[85], and regions for fine-mapping risk loci were defined by taking a 1 Mb window around the lead SNP of each locus. Upon fine-mapping each region, 95% credible sets were defined, and posterior inclusion probabilities (PIP) per SNP in the fine-mapping region were computed to evaluate variant causality.

To nominate the top candidate causal gene at each locus, we employed the combined SNP-to-gene linking strategy elaborated by Gazal et al.[49], integrating genome annotations (i.e., exonic, promoter regions) and functional annotations (i.e., fine-mapped cis-eQTL, regulatory annotations) with the putatively causal variants in each locus determined by fine-mapping. We were able to apply this SNP-to-gene method on the 50 fine-mapped loci, which contained a credible set with at least one SNP with PIP > 0.1. Twenty-three of these 50 loci resulted in a predicted causal SNP-gene pair, and the remaining 27 loci were assigned the nearest gene to the SNP with the highest PIP.

## LOF burden tests

Results from gene-level burden tests of predicted LOF protein-coding variants (pLOF mask as defined by LOFTEE[86]) in nominated genes of interest with TG/HDL among the exome-sequenced individuals in the UK Biobank[50] ($n$ = 343,470 with TG/HDL measurements available) was extracted from https://app.genebass.org/ (Accessed: June 16, 2022). For each gene, we filtered the phenotype to "ratio TG HDL custom," set the burden set to "pLOF," set the burden testing method to "burden," and extracted the resulting $p$-value and effect size.

To perform LOF burden tests for *PLA2G12A* in the UK Biobank exome sequenced individuals, we used different definitions for LOF variants (masks) with increasing stringency to classify variants. The most stringent mask included disruptive variants, as described in the previous paragraph. In subsequent masks, additional variants were included using the following five computational prediction tools: Polyphen2_HDIV[87], Polyphen2_HVAR[87], SIFT[88], LRT[89], and MutationTaster[90]. "High confidence LOF variants" included variants predicted deleterious by all five tools, "Moderate confidence LOF variants" included rare (MAF < 1%) variants predicted deleterious by at least 1 tool, and the least stringent mask included all rare variants (MAF < 1%).

Gene-based burden tests were performed using REGENIE as detailed in the documentation (https://rgcgithub.github.io/regenie/options/#gene-based-testing). In brief, variants in *PLA2G12A* were extracted from the helper file ukb23149_450k_OQFE.sets.txt.gz using the "swiss-army-knife" tool from the UK Biobank Research Analysis Platform (https://dnanexus.gitbook.io/uk-biobank-rap). A site VCF was generated, and variants were annotated with Variant Effect Predictor (r110.1)[53] for mask filtering downstream. The interim BGEN release ("Population level exome OQFE variants, BGEN format - interim 450k release") was converted to PGEN format to produce a filtered dataset. We retained samples with a sample call rate greater than 95% and with heterozygosity values less than the median + 3x the interquartile range (IQR) and excluded those with reported gender-genetic sex mismatch. Variants with genotype call rate <98%, MAF < 0.01, HWE $p < 1 \times 10^{-8}$, and minor allele count <100 were excluded to create a genome-wide representative dataset for Step 1 (Ridge Regression) of REGENIE v3.1.2. TG, HDL, age, sex, and the first 20 principal components of ancestry were extracted using the "table-exporter" from the UK Biobank Research Analysis Platform. The TG/HDL ratio was natural log-transformed, and the covariates included were age, age², sex, and the first 20 principal components of ancestry for statistical models. A total of 396,335 samples were included in the analysis. In Step 2 of REGENIE, a rank inverse normal transformation was applied on the phenotype and a burden test ("maximum method" in REGENIE) was conducted on variants included in 4 masks of increasing deleteriousness. We also performed the same model after conditioning upon as well removing the two missense variants rs114816312 and rs41278045.

## Assessment of directional effects using functional genomic annotations of putative causal SNPs

To annotate the direction-of-effect of genes on TG/HDL, we examined putative causal fine-mapped variants (PIP > 0.1), which were annotated to be exonic or fine-mapped eQTLs for consequences on protein function or gene expression, respectively. For causal eQTLs, the direction-of-effect was inferred based on concordance of the variant's effect on gene expression (eQTL beta) and the effect on TG/HDL (GWAS beta) in the respective study in which the eQTL was fine-mapped for (GTEx[60], eQTLGen[91]). For example, a positive eQTL beta and GWAS beta would indicate that an increase in gene expression would increase TG/HDL. Similarly, the functional consequence of each putative causal coding variant on protein function was predicted using established computational tools PolyPhen-2[87] and SIFT[88] as determined by Ensembl Variant Effect Predictor (VEP)[53] and then compared with the direction of effect of the variant on TG/HDL (GWAS beta).

## Associations of heritable gene expression with TG/HDL in the UK Biobank

To examine the effect of heritable gene expression on TG/HDL for the top nominated genes of interest in relevant tissues (sub-cutaneous and visceral adipose, liver, and skeletal muscle), we first assigned per-tissue gene expression scores to each UK Biobank participant using established methods (Functional Summary-based Imputation [FUSION])[51]. In brief, gene expression reference weights from the Genotype-Tissue Expression (GTEx) Project v8 for tissues of interest were extracted from the FUSION archive (http://gusevlab.org/projects/fusion/#gtex-v8-multi- tissue-expression). The expression weights per gene and their corresponding imputed genotypes in the UK Biobank genotyped population were combined into a predicted gene expression score for each individual by computing a linear combination across the SNPs expression weights and genotypes, then dividing by the number of non-missing SNPs as implemented in the Plink score utility[92]. TG/HDL measurements were normalized as described above and regressed onto predicted gene expression scores as done previously[93].

## Association of heritable *PLA2G6* muscle expression on metabolic outcomes

Using the same *PLA2G6* muscle expression weights as described previously, we investigated the role of predicted *PLA2G6* muscle expression on metabolic outcomes T2D[41], FI[14], CVD[47], and NAFLD[40] using the FUSION association test function (FUSION.assoc_test) with default parameters[51].

## Differential expression of *PLA2G6* in skeletal muscle upon TZD treatment

Skeletal muscle biopsies were obtained from individuals before and after 3 months of TZD treatment ($n$ = 35)[20]. Total RNA was extracted (Qiagen #74104) and sequenced (Illumina TruSeq) according to the manufacturer's protocol to a depth of at least 30 million reads per sample with paired-end 100 bp reads. Raw reads were aligned using Kallisto[94] with default parameters and normalized for library size to generate counts per million (CPM) mapped reads. Differential expression analysis for *PLA2G6* upon TZD treatment was performed in dream (variancePartition v1.20.0, edgeR v3.32.1)[95] with pre/post TZD treatment as the fixed effect and patient ID as a random effect to account for paired samples.

## Correlations between adipose tissue and skeletal muscle *VGLL3* expression and glucose disposal rate in human participants

Subcutaneous adipose tissue and skeletal muscle biopsies were obtained and RNA-sequenced, and clinical measurements of insulin sensitivity along with other metabolic health metrics were assessed in a clinical cohort ($n$ = 35) as previously described[20]. Total RNA was extracted (Qiagen #74104) and sequenced (Illumina TruSeq) according to the manufacturer's protocol to a depth of at least 30 million reads per sample with paired-end 100 bp reads. Raw reads were aligned using Kallisto[94] with default parameters to generate gene counts and normalized for library size to generate counts per million (CPM) mapped reads. *VGLL3* muscle and adipose expression levels were regressed on glucose disposal rate measured by hyperinsulinemic-euglycemic clamp in the 35 individuals using linear regression (lm function, R stats v3.5.1). To avoid confounding effects as a result of insulin-sensitizing treatment tested in this clinical test, only data from tissue biopsies obtained before treatment were used in this analysis. Mediation analysis to quantify the proportion of effect of *VGLL3* expression on Rd mediated by BMI was conducted using the mediate function as implemented in R package mediation v4.5.0[96] using the following linear models: (1) Rd ~ *VGLL3* expression + BMI, and (2) *VGLL3* expression ~ BMI.

**RNA-sequencing of human adipocytes through differentiation**

Human Simpson-Golabi-Behmel syndrome (SGBS) preadipocyte cells were differentiated as we have previously described[66]. Total RNA was extracted (Qiagen #74104) from cells at days 0, 4, 8, and 15 and sequenced (Illumina TruSeq) according to the manufacturer's protocol to a depth of at least 50 million reads per sample with paired-end 100 bp reads. Raw reads were aligned using Kallisto[94] with default parameters to generate gene counts. Genes with a count greater than 15 in all replicates of any differentiation day were retained for analysis. Following normalization for library size, differential expression analysis for *TNFAIP8* was conducted using a linear mixed model with differentiation days 0 vs. 4 as a fixed effect as instantiated in dream (variancePartition v1.20.0, edgeR v3.32.1)[95].

**Reporting summary**

Further information on research design is available in the Nature Portfolio Reporting Summary linked to this article.

# Data availability

Full GWAS summary statistics from this study are available at the GWAS Catalog (https://www.ebi.ac.uk/gwas/) under study accession codes GCST90435481, GCST90435482, GCST90435483. Individual-level genomic and phenotypic data from the UK Biobank are available to researchers upon application (https://ukbiobank.ac.uk). The RNA-seq data for Human SGBS preadipocyte cells from this study are available at the GEO under accession code GSE274839. The raw RNA-seq data for skeletal muscle and subcutaneous adipose tissue biopsies before and after TZD treatment cannot be made available to the public due to lack of patient consent. They will be made available from the corresponding author upon request under a Data Transfer Agreement (DTA) and transferred via FTP when the DTA is complete. The summary statistics of previous insulin resistance-related GWAS used in this study are available as described in Supplementary Data 3. The functional genomic annotations for SNP to gene linking used in this study were downloaded (November 2023) from https://alkesgroup.broadinstitute.org/cS2G. The Genebass exome-based association statistics in the UK Biobank (accessed November 2023) are available here: https://app.genebass.org/. The Functional Summary-based Imputation (FUSION) GTEx v8 multi-tissue expression statistics used in this study were downloaded (November 2023) from http://gusevlab.org/projects/fusion/#gtex-v8-multi-tissue-expression. The metabolic disease/trait associations meta-analyzed in this study (accessed November 2023) were obtained from the Common Metabolic Diseases Knowledge Portal (cmdkp.org) https://hugeamp.org/variant.html?variant=rs1045241. The ChIP-seq of *PPARG* in human adipose stem cells[67] and estrogen receptor (*ESR1*) in human breast cell line ZR-75-1[69] used in this study were downloaded from the GEO under accession codes GSM534493 and GSM798427, respectively.

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

## Acknowledgements

This work was supported by grants from the National Institute of Diabetes and Digestive and Kidney Diseases (1R01DK123422 to A.R.M. and 1R01DK125490 to J.F. and A.R.M.), the National Heart, Lung, and Blood Institute (1R01HL159760 to A.R.M.), and a Ruth L. Kirschstein Institutional National Research Service Award T32 GM008666 from the National Institute of General Medical Sciences (to N.D.). G.M.P. and P.N. received support from grants R01HL142711 and R01HL127564. We would like to thank Jerrold M. Olefsky for helpful discussions. We would like to thank the participants of the UK Biobank for donating their data for scientific investigation. Analysis of UK Biobank data was conducted under application number 51436 and 41189.

## Author contributions

N.D., G.M.P., and A.R.M. designed the study. N.D., Y.W. Z.Z., J.S.D., R.K., J.F., generated/acquired and analyzed the data. N.D., Y.W. Z.Z., T.A., G.M.P., and A.R.M., were involved in data interpretation. N.D. and A.R.M. drafted the manuscript. N.D., Y.W., J.S.D., P.N., J.F., T.A., G.M.P., and A.R.M. were involved in critical manuscript revision.

## Competing interests

P.N. reports research grants from Allelica, Apple, Amgen, Boston Scientific, Genentech/Roche, and Novartis, personal fees from Allelica, Apple, AstraZeneca, Blackstone Life Sciences, Creative Education Concepts, CRISPR Therapeutics, Eli Lilly & Co, Foresite Labs, Genentech/Roche, GV, HeartFlow, Magnet Biomedicine, and Novartis, scientific advisory board membership of Esperion Therapeutics, Preciseli, and TenSixteen Bio, scientific co-founder of TenSixteen Bio, equity in MyOme, Preciseli, and TenSixteen Bio, and spousal employment at Vertex Pharmaceuticals, all unrelated to the present work. The remaining authors declare no competing interests.
