## [Peer Review File · Nature Communications]

Genome-wide discovery and integrative genomic characterization of insulin resistance markers using serum triglycerides to HDL-cholesterol ratioREVIEWER COMMENTS

Reviewer #1 (Remarks to the Author):

Please note that in addition to the comments raised below, publication of a paper on the exact same topic does mean that the novelty of this work is compromised. My suggestions below were written before having knowledge of this publication and are hopefully still helpful for the authors.

Insulin resistance is difficult to measure at biobank scale due to the lack of direct measures of insulin resistance (e.g., during hyperglycemic-euglycaemic clamps) in these large studies. Therefore, surrogate measures of insulin resistance have been developed including triglyceride to HDL ratio (TG/HDL), which is the focus of this manuscript from DeForest and colleagues.

DeForest and colleagues present a biobank-scale GWAS of TG/HDL, a proxy measure of insulin resistance, conducted in the UK Biobank study. This included 382,129 in sex combined analyses and identified 251 TG/HDL associated genetic loci, 17 of which showed evidence of sex dimorphic associations in sex stratified GWAS within the same cohort. Replication analysis was conducted in Mass General Biobank (N=37,545) which has ~10% sample size of the discovery GWAS in UK Biobank. The authors further prioritised loci through the identification of those with "boosted" P values of association with TG/HDL compared to TG or HDL alone, proposing these could be more relevant to insulin resistance rather than lipid regulation. 62 of the 251 loci were prioritised and fine-mapping was conducted to prioritise putative causal variants at 57 of these loci. These causal variants were used to prioritise candidate causal genes. Gene-level follow up focused in on 4 genes providing some tissue specific mechanistic hypotheses for the role of these candidates in insulin resistance. PLA2G12A, PLA2G6 and VGLL3 were prioritised in sex combined analyses and TNFAIP8, which was identified as the candidate causal gene at a locus that displays sex dimorphic effects.

Overall, the text is well written, and the figures well presented. However, there are areas which lack detail and therefore do not allow a full picture of the analyses to be obtained. In addition, some expansion of analyses and additional analyses may allow better support for some of the conclusions drawn in this manuscript, which have been included below for consideration.

Major comments

1) Circulating triglyceride levels are not comparable in the fasting vs non-fasted states. This may reduce the transferability and suitability of TG/HDL ratio as a proxy measure of insulin resistance. Can you confirm whether the correlation analyses in the 45 individuals with clamp measures was done using TG and HDL measures in the fasted or non fasted state? UK Biobank measures of triglycerides and HDL have been conducted on non-fasted samples and therefore will be impacted by the variability of triglyceride levels in the non-fasted state.

2) For the observational correlation analyses between TG/HDL and other measures of in the 45 individuals who underwent hyperinsulinemic-euglycemic clamp measures some more information would be helpful to fully interpret the results (Lines 75-82). Reference 16 which refers to Sears et al (2009, PNAS) outlines these individuals were insulin sensitive and resistant and that these individuals undertook a hyperinsulinemic-euglycemic clamp before and after TZD treatment. It is not clear from the current description of the relevant methods (lines 521-527) what treatment group or insulin sensitivity class these 45 individuals were from both of which could impact the individual measures compared as well as their pairwise correlations. It would be helpful if authors could clarify this in the text. If there are different groups within the subset of this cohort included in the correlation, it would be informative to demonstrate how this impacts the correlations as appropriate.

3) It is not explicitly stated in the manuscript whether genetic ancestry was accounted for in the

GWAS analysis (e.g. only including individuals of European ancestry; lines 83-86). Stratifying by genetic ancestry group is usually done in GWAS analyses. Please provide clarification of this.

4) Genetic correlation analyses via LDSC for TG/HDL with other relevant traits focused on continuous traits HOMA-IR, fasting insulin, fasting glucose and WHR adjusted for BMI and diseases: T2D, NAFLD, CVD. The addition of other insulin resistance and other glycemic traits in the genetic correlations could allow more complete interpretation of the results for example there are other traits in GWAS from the MAGIC consortium beyond those used at present that reflect insulin resistance e.g., 2hr glucose (Chen et al 2021, Nat Gen) and other indices of insulin resistance e.g. Modified Stumvoll Insulin Sensitivity Index (Williamson et al 2023, Nat Gen). For glycemic traits Chen et al (2021) also present results for HbA1C. Other glycemic traits reflecting insulin secretion e.g. HOMA-B (Manning et al 2012, Nature Genetics) could also be considered to provide evidence of specificity of TG/HDL to insulin resistance. Please also provide more details of the exact summary statistics used as input in the methods (Lines 590-600). Many of these GWAS studies provide multi-ancestry and ancestry specific results as well as GWAS conducted for traits adjusted and unadjusted for BMI. The overall genetic correlation with triglycerides and HDL alone could also be informative to demonstrate the how unique the genetic architecture of this TG/HDL is from the measures it is composed of.

5) Genetic colocalisation would allow more formal testing of the sharedness of the genetic signals between TG/HDL and other insulin resistance traits described in Supplementary table 3 and at lines 165-173. The LD between with lead TG/HDL associated SNP at a given locus and the overlapping SNP for the other traits of interest would also be informative, as it is possible that the signals for each trait could be independent despite being at the same distance-defined locus.

6) Gene burden analysis methods and results description is unclear (line 628-633). Can you provide some more detail regarding the Genebase look up before doing individual level LOF burden tests for PLA2G12A using a different approach the other prioritised genes using Regenie? The Genebase resource uses different variant masks and a different analysis method (SAIGE-GENE) than the PLA2G12A later presented. Are the results consistent using Regenie across the prioritised gene list in your analyses?

More detail is also needed for the sequencing QC in the individual level data analysis in UK Biobank e.g. the sample size included (lines 633-644). The summary statistics including standard errors and number of variant carriers included should be provided for PLA2G12A for all burden masks tested.

7) In general, increased detail in the methods would aid ease of readability and reproducibility. This is particularly the case in the following sections, in addition to those mentioned above:

7a. Line 535-536 – Please briefly describe the GLGC guidelines that were used to filter the samples. Based on the referred guidelines, clarification as to what data fields were used to determine lipid lowering medication use in UK Biobank is also needed.

7b. Line 546 – Please specify what reference panel was used for imputation.

7c. Lines 565-572 – a brief description of the Mass General Biobank would be helpful (e.g., recruitment, age, etc), as has been done for UK Biobank.

7d. RNA-sequencing of tissue samples, lines 680-701. More detail is needed to give a clearer view on this analysis as well as an overview of sample processing and data generation e.g., RNA extraction, sequencing, data QC, processing, and statistical analysis. Whilst microarray expression analysis is mentioned in reference 84 (Sears, 2009; PNAS – also reference 16) RNA sequencing is not mentioned. This is also the case for SGBS cells at lines 704-709.

8) In general, more explicit disclosure throughout of what results were obtained from novel analyses compared previously published publicly available datasets would be helpful. Sometimes this is hard to deduce in the current manuscript. An additional section in the methods outlining the exact data sources used would be helpful for reproducibility.

Minor comments:

9) Pearson's correlation is stated to have been used in line 81 whereas Spearman's correlation is stated in SF1 (p47) and methods (L547), is this a typo? Please clarify the approach taken in the text.

10) Please highlight which of the 251 loci identified are novel for insulin resistance, and other glycaemic traits.

11) Sex specific GWAS analyses focus on examining the sex dimorphic effect of the 251 loci identified for TG/HDL in sex combined analyses, using the same analytical approach. Whilst the comparison of these loci is included, it would be informative describe these results more fully. Particularly to state if any additional genome wide significant loci were identified in the sex stratified analyses on top of those identified in the sex combined GWAS.

12) Alongside the measures of fat distribution (e.g. VAT adjBMI), it may also be helpful to include BMI and WHR unadjusted for BMI allow a comparison of effects on overall adiposity with fat distribution using look ups of the BMI adjusted results for the MRI-based measures included in Agrawal et al (2022). Indicating more clearly what summary statistics were obtained from BMI adjusted GWAS analyses (e.g. WHR, ASAT) would aid readability (Figure 5, lines 350, 355-366)

13) Supplementary table 3 and Figure 2: In Supplementary Table 3 the SNP, beta, se and p value provided for look up of Fasting insulin and HOMA-IR are combined. Please split these into individual traits so it is clear what trait the summary statistics for each variant are referring to.

Please define what is classed as no_overlap. Does this refer to those that do not meet genome wide significance threshold of $P < 5e-8$?

It would be useful to see which loci have consistent expected direction of effects at a more relaxed significance threshold than that included at present.

Reviewer #2 (Remarks to the Author):

In this manuscript the authors used the UK biobank dataset (N= 382,129) to discover loci associated with TG/HDL ratio, a marker of insulin resistance. The authors found 251 independent loci, of which 62 showed stronger association with TG/HDL ratio than with either of the measures, TG or HDL. Using additional fine-mapping approaches and other datasets, the authors go onto to describe some loci in more detail.

This is a really nice piece of work which is clearly written and easy to follow.

However, I do have some general comments/concerns:

1. Given the just published paper but Oliveri et al., 2024, it would be nice to understand a bit better how the present results compare with those of the Oliveri et al. especially seeing as the underlying data are the same. Both this study and the Oliveri et al 2024 use the same TG/HDL ratio and the same discovery dataset (UK biobank). The sample sizes and software packages used differ, and the number of resulting independent loci also differ. It would be helpful (and reassuring to the reader) to try and find out the overlap in the findings and the level of concordance in the results.

2. This present work has a different emphasis in the post-GWAS analysis, so I don't believe the other publication detracts from the findings described here. I just feel that given that the two discovery analyses are so similar it would help the reader to better understand the differences and also the consistency, or lack thereof, of the overall findings.

3. I found it particularly perplexing that some of the top novel loci discussed and analysed in detail here, I could not see in the supplementary tables of Oliveri et al. e.g. PLA2G12A, PLA2G6 and VLL. It could be that the index variants were different, or that I missed them but it would be good to understand the key similarities and differences in the results obtained.

4. When comparing the genetic correlations of TG/HDL ratio with some other measures and diseases, the authors compare the genetic correlation of TG/HDL with that of other glycaemic traits with the same traits. It wasn't always clear to me why some statements were made, as the measures (without a statistic to test for differences) looked broadly similar by eye. For example, the authors state "Positive genetic correlations were observed between TG/HDL and T2D ($r_g=0.51$, $p=1.9 \times 10^{-36}$), CVD ($r_g=0.31$, $p=2.8 \times 10^{-28}$), and NAFLD ($r_g=0.74$, $p=0.001$). Notably, TG/HDL had comparable genetic correlation with T2D as compared to glycaemic traits (FI $r_g=0.6$, $p=5.2 \times 10^{-29}$; FG 131 $r_g=0.57$, $p=3.8 \times 10^{-18}$) but greater genetic correlation with CVD (FI $r_g=0.29$, $p=3 \times 10^{-9}$; FG $r_g=0.12$, $p=1 \times 10^{-4}$) and NAFLD (FI $r_g=0.9$, $p=0.01$; FG $r_g=0.3$, $p > 0.05$)." So the authors claim that the genetic correlation of TG/HDL with T2D $r_g=0.51$ is comparable to that of glycaemic traits (FI $r_g=0.6$; FG $r_g=0.57$); but then claim they see "greater genetic correlation with CVD". However, their data show TG/HDL $r_g=0.31$ with CVD vs $r_g=0.29$ for FI and CVD. I'd say this is pretty comparable too? Same for NAFLD TG/HDL $r_g=0.74$, while FI is $r_g=0.9$? In fact, for the latter FI looks more correlated with NAFLD.

5. For the boosted score the authors compute, it would be helpful to be clear in the methods whether the $-\log_{10}$ p-value for all three traits being compared TG/HDL, TG and HDL came from association testing of the exact same samples so there are no underlying differences in sample sizes between the traits.

6. In the fine-mapping section it would be helpful to know what made loci not mappable?

7. For the nominated gene PLA2G12A, the authors show the variant rs114816312 has $\beta=0.1$, this particular variant is predicted to lead to a missense, p.D111N. A second variant, rs41278045 is also predicted to lead to a missense, p.C131R, and both are predicted to be deleterious. Still, when performing gene-burden tests with predicted LOF variants the effect size is one order of magnitude smaller 0.01 (mentioned in the text and shown in ST5) but Figure 4B shows a different effect size. Do the authors have a proposed mechanism for how missense variants will affect the TG/HDL ratio a lot more than a burden of LOF variants, when the authors propose that in both situations the missense variants are impacting protein function? Another question I have here is whether the burden of LOF effects changes when conditioned on rs114816312 and rs41278045 variants? The authors removed the two variants (rs114816312 and rs41278045) from the burden but if there is LD between either of them and some of the other variants in the burden could this be the reason for the burden effects shown?

8. At the TNFAIP8 locus the authors refer to serum proteomics data from a different study, have the authors looked directly at the proteomics data available in biobank?

9. Overall, in terms of nominating causal genes, I think the authors need to express a little more caution as the data shown are not definitive. Namely, the authors focused their analysis on variants that had the higher PIP in their credible set but in some examples this was still not particularly strong (for example, $PIP=0.1$ in PLA2G6 and also see concerns about the LOF burden at PLA2G12A), and the authors have not shown that those specific variants definitely affect protein function/expression in an allele-specific manner. Nor have they shown a direct link between protein function and the measure they are looking at (TG/HDL ratio). So there is still room for other variants (and genes) at these loci to be implicated. I'd suggest that they propose these as candidate causal as they have shown correlative data, but not mechanistic data directly linking specific variants to gene expression or protein expression/function.

10. I also think the authors need to be more cautious regarding nomenclature as their "causal variants", should more correctly be called "putative causal variants". For most of the results shown, the 95CS still had multiple variants, and the authors focused on those with highest PIP values (and as mentioned above $PIP=0.1$ is not that high) leaving room for other variants to be causal, or also causal. In addition, any statistical fine-mapping approach is limited by the SNPs present in the data (the true causal variant may not actually be in there but just others tagging it) so it is not possible to be certain the best PIP variants are indeed causal, especially in the absence of functional data directly modelling effect of that variant.

11. The below is a minor point but in the discussion the authors cannot claim to have the largest GWAS for TG/HDL ratio given the recently published paper which actually had a marginally larger sample size of 402,398. There are few places in the discussion where this now needs to be updated

and probably refer to this other published paper at least by stating while this paper was under review this other one was published?

Reviewer #3 (Remarks to the Author):

The rationale of this paper is questionable as this reviewer was not aware of a way to measure insulin sensitivity in general populations using lipids values. Reading the ref 15 paper (only paper referred to justify this study) it is said : "The optimal triglyceride/high-density lipoprotein cholesterol ratio for predicting insulin resistance and LDL phenotype was 3.5 mg/dl; a value that identified insulin-resistant patients with a sensitivity and specificity comparable to the criteria currently proposed to diagnose the metabolic syndrome. The sensitivity and specificity were even greater for identification of patients with small, dense, LDL particles. In conclusion, a plasma triglyceride/high-density lipoprotein cholesterol concentration ratio $>$ or $=3.5$ provides a simple means of identifying insulin-resistant, dyslipidemic patients who are likely to be at increased risk of cardiovascular disease." It is obvious that this old study (never reproduced) was intended to determine most insulin resistant subjects among high risk of CV diseases people, with a threshold for IR, but not to have a continuous measurement of insulin sensitivity in general population. Therefore this reviewer expressed concern that all the current study is flawed. This feeling is reinforced by the so called "replication" from an old study (ref 16, with only 45 subjects, which did not show this correlation). The supp figure 1 clearly shows that at medium (most frequent in general population) Trig/HDL ratio the clamp measures insulin sensitivity is extremely variable, showing again that this analysis is probably valuable for a fragment of at risk for CV patients but not as an epidemiology tool.

The GWAS analysis for Trig/HDL ratio is standard, with no obvious replication in a medium size American cohort. The correlation with previous GWAS for T2D etc was expected due to the abnormal lipid values in these insulin resistant populations.

This reviewer is not convinced that the "boosted priority TG/HDL association study for specific IR genes" has any physiology meaning, other than being a statistical game. The "boost score" is for this reviewer artificial and in this reviewer's memory LPL is not a T2D associated gene.

The search for "causative" genes is interesting but it is apparently only based on silico data from databases and previous papers with no novel and original mechanistic proof of concept that any of the "new" genes is physiologically associated with both insulin sensitivity and lipids metabolism.

BTW the conclusion of the paper shows what remains to be done, from the current data, enrich our knowledge of the physiology of the crosstalk between lipid and glucose metabolism.

all_Reviewer's Comments:

Reviewer #1 (Remarks to the Author)

Please note that in addition to the comments raised below, publication of a paper on the exact same topic does mean that the novelty of this work is compromised. My suggestions below were written before having knowledge of this publication and are hopefully still helpful for the authors.

We understand the reviewer's concerns and appreciate all the suggestions. Despite using the same TG/HDL measurements and discovery cohort as the recent study (PMID: 38200128), our post-GWAS emphasis differs significantly. Through providing tissue-specific mechanistic hypotheses, our study aimed to identify novel insulin resistance genes by prioritizing candidate causal genes through fine-mapping and computing "boosted" scores. The other study focused on using TG/HDL associated loci to annotate tissues, cell types, and pathways relevant to the physiology of insulin resistance. Furthermore, our methodological approaches differ: we included variants such as insertions/deletions and those on the sex chromosomes, and took a more liberal threshold of minor allele frequency (MAF > 0.001) to identify independent loci. Using this approach, we additionally identified and characterized 2 novel putative insulin resistance genes (PLA2G12A and PLA2G6).

Insulin resistance is difficult to measure at biobank scale due to the lack of direct measures of insulin resistance (e.g., during hyperglycemic-euglycaemic clamps) in these large studies. Therefore, surrogate measures of insulin resistance have been developed including triglyceride to HDL ratio (TG/HDL), which is the focus of this manuscript from DeForest and colleagues.

DeForest and colleagues present a biobank-scale GWAS of TG/HDL, a proxy measure of insulin resistance, conducted in the UK Biobank study. This included 382,129 in sex combined analyses and identified 251 TG/HDL associated genetic loci, 17 of which showed evidence of sex dimorphic associations in sex stratified GWAS within the same cohort. Replication analysis was conducted in Mass General Biobank (N=37,545) which has ~10% sample size of the discovery GWAS in UK Biobank. The authors further prioritised loci through the identification of those with "boosted" P values of association with TG/HDL compared to TG or HDL alone, proposing these could be more relevant to insulin resistance rather than lipid regulation. 62 of the 251 loci were prioritised and fine-mapping was conducted to prioritise putative causal variants at 57 of these loci. These causal variants were used to prioritise candidate causal genes. Gene-level follow up focused in on 4 genes providing some tissue specific mechanistic hypotheses for the role of these candidates in insulin resistance. PLA2G12A, PLA2G6 and VGLL3 were prioritised in sex combined analyses and TNFAIP8, which was identified as the candidate causal gene at a locus that displays sex dimorphic effects.

Overall, the text is well written, and the figures well presented. However, there are areas which lack detail and therefore do not allow a full picture of the analyses to be obtained. In addition,

some expansion of analyses and additional analyses may allow better support for some of the conclusions drawn in this manuscript, which have been included below for consideration.

Major comments

1) Circulating triglyceride levels are not comparable in the fasting vs non-fasted states. This may reduce the transferability and suitability of TG/HDL ratio as a proxy measure of insulin resistance. Can you confirm whether the correlation analyses in the 45 individuals with clamp measures was done using TG and HDL measures in the fasted or non fasted state? UK Biobank measures of triglycerides and HDL have been conducted on non-fasted samples and therefore will be impacted by the variability of triglyceride levels in the non-fasted state.

The surrogate markers of insulin resistance including TG and HDL in the 45 individuals with glycemic clamp data were performed in the fasting state pre-glycemic clamp. We thank the reviewer for pointing this out and have added the following to the methods section: “We included participants across all insulin sensitivity classes and only included biological samples from before TZD treatment **in the fasting state, pre-glycemic clamp.**”

2) For the observational correlation analyses between TG/HDL and other measures of in the 45 individuals who underwent hyperinsulinemic-euglycemic clamp measures some more information would be helpful to fully interpret the results (Lines 75-82).

Reference 16 which refers to Sears et al (2009, PNAS) outlines these individuals were insulin sensitive and resistant and that these individuals undertook a hyperinsulinemic-euglycemic clamp before and after TZD treatment. It is not clear from the current description of the relevant methods (lines 521-527) what treatment group or insulin sensitivity class these 45 individuals were from both of which could impact the individual measures compared as well as their pairwise correlations. It would be helpful if authors could clarify this in the text.

If there are different groups within the subset of this cohort included in the correlation, it would be informative to demonstrate how this impacts the correlations as appropriate.

We agree that this needs to be clarified; we added the following statement to the methods section: “**We included participants across all insulin sensitivity classes and only included biological samples from before TZD treatment** in the fasting state, pre-glycemic clamp.” As the reviewer suggested we have highlighted in SF1A (originally SF1) the group membership of participants in insulin sensitive and resistant classes. However, there were so few in the “non-resistant” category (n = 10) that a subcohort correlation analysis would not be statistically robust.

3) It is not explicitly stated in the manuscript whether genetic ancestry was accounted for in the GWAS analysis (e.g. only including individuals of European ancestry; lines 83-86). Stratifying by genetic ancestry group is usually done in GWAS analyses. Please provide clarification of this.

We accounted for genetic similarity/ population stratification by including the first 20 principal components of genetic ancestry in accordance with the best practices recommended by The

National Academies of Sciences, Engineering, and Medicine (NASEM) (ISBN: 978-0-309-70068-9). This was originally described in the methods section and to make it clear to potential readers we have added it to the results as well (line 91). We note that the effect sizes (betas) from the European-only analysis and the full multi-ancestry analysis were very strongly correlated ($r = 0.913$) and furthermore, the 251 SNPs/loci that exceeded the genome-wide significance threshold showed almost perfect correlation ($r = 0.9995$).

4) Genetic correlation analyses via LDSC for TG/HDL with other relevant traits focused on continuous traits HOMA-IR, fasting insulin, fasting glucose and WHR adjusted for BMI and diseases: T2D, NAFLD, CVD. The addition of other insulin resistance and other glycemic traits in the genetic correlations could allow more complete interpretation of the results for example there are other traits in GWAS from the MAGIC consortium beyond those used at present that reflect insulin resistance e.g., 2hr glucose (Chen et al 2021, Nat Gen) and other indices of insulin resistance e.g. Modified Stumvoll Insulin Sensitivity Index (Williamson et al 2023, Nat Gen). For glycemic traits Chen et al (2021) also present results for HbA1C. Other glycemic traits reflecting insulin secretion e.g. HOMA-B (Manning et al 2012, Nature Genetics) could also be considered to provide evidence of specificity of TG/HDL to insulin resistance. Please also provide more details of the exact summary statistics used as input in the methods (Lines 590-600). Many of these GWAS studies provide multi-ancestry and ancestry specific results as well as GWAS conducted for traits adjusted and unadjusted for BMI. The overall genetic correlation with triglycerides and HDL alone could also be informative to demonstrate the how unique the genetic architecture of this TG/HDL is from the measures it is composed of.

The reviewer's point is well taken; we performed additional analyses computing pairwise genetic correlations including the following traits:

- 2hr glucose (Chen et al 2021, Nat Gen)
- HbA1c (Chen et al 2021, Nat Gen)
- Modified Stumvoll Insulin Sensitivity Index (ISI) (Williamson et al 2023, Nat Gen)
- HOMA-B (Manning et al 2012, Nature Gen)
- TG and HDL individually

These additional analyses revealed a very strong genetic correlation between TG/HDL with TG ($r_g=0.96$) and HDL ($r_g=-0.82$) suggesting a large degree of sharing but not complete overlap of genetic architectures. The magnitude of genetic correlation between TG/HDL and HbA1c ($r_g=0.12$) or 2hr glucose ($r_g=0.23$) are comparable to FG ($r_g=0.17$) and smaller than genetic correlation between TG/HDL and FI ($r_g=0.49$), HOMA-IR ($r_g=0.49$) or ISI ($r_g=-0.47$). Even though HOMA-B and HOMA-IR are calculated from the same fasting measurements, HOMA-IR ($r_g=0.49$, $p=4.11 \times 10^{-10}$) had a stronger and more significant correlation with TG/HDL than HOMA-B ($r_g=0.41$, $p=4.78 \times 10^{-7}$). Overall, these analyses show that TG/HDL is more genetically correlated with measures of insulin resistance than glycemia or insulin secretion.

Where possible we utilized BMI adjusted summary statistics and restricted to European ancestry for calculations of genetic correlation. The additional genetic correlations and information regarding the input GWAS summary statistics have been added to the results (lines 125-142) and incorporated into ST3 (originally ST2).

5) Genetic colocalisation would allow more formal testing of the sharedness of the genetic signals between TG/HDL and other insulin resistance traits described in Supplementary table 3 and at lines 165-173. The LD between with lead TG/HDL associated SNP at a given locus and the overlapping SNP for the other traits of interest would also be informative, as it is possible that the signals for each trait could be independent despite being at the same distance-defined locus.

We agree with the reviewer's point and performed formal genetic colocalization testing for TG/HDL and T2D, WHR, CVD, and FI. HOMA-IR was not included because the summary statistics were derived from a much older study (PMID: 20081858) with fewer samples ($n=37,037$) and utilized a less stringent significance threshold ($P\text{-value} < 2 \times 10^{-5}$). We computed posterior probabilities of genetic colocalization between the 251 TG/HDL associated loci and T2D, WHR, CVD, and FI using COLOC (PMID: 24830394). These are incorporated as a new ST5 (including R^2 between non-identical lead SNPs) and detailed in the revised Methods.

Overall, these analyses show that the large majority of positionally overlapping loci (i.e. lead variant from Trait X falls within genomic coordinates of a TG/HDL locus boundary) contain shared genetic signals. Applying a posterior probability threshold (i.e. $PP.H3.abf + PP.H4.abf \geq 0.99$) previously used to evidentiate overlapping genetic signals (PMID: 25743184) we found that of the 93 positional overlaps between TG/HDL and WHR (Figure 2B), 86 formally overlapped (55/61 for T2D, 36/42 for CVD, and 24/27 for FI). We have noted these findings in the Results (lines 183-185) and highlighted overlapping signals in a new SF4 alongside the full colocalization results presented in ST5.

6) Gene burden analysis methods and results description is unclear (line 628-633). Can you provide some more detail regarding the Genebass look up before doing individual level LOF burden tests for PLA2G12A using a different approach the other prioritised genes using Regenie? The Genebass resource uses different variant masks and a different analysis method (SAIGE-GENE) than the PLA2G12A later presented. Are the results consistent using Regenie across the prioritised gene list in your analyses?

More detail is also needed for the sequencing QC in the individual level data analysis in UK Biobank e.g. the sample size included (lines 633-644). The summary statistics including standard errors and number of variant carriers included should be provided for PLA2G12A for all burden masks tested.

As the reviewer requested we have provided more details on the Genebass lookup and this is incorporated into the methods (lines 698-700). For each gene we filtered the phenotype to "ratio TG HDL custom", set the burden set to "pLOF" and set the burden testing method to "burden" and extracted the resulting p-value and effect size.

We performed our own burden testing for PLA2G12A to dissect the effects of removing the lead GWAS variants (rs114816312 and rs41278045, figure 4B) and altering mask stringency for variants included. As the reviewer requested we have added a new ST9 that shows the summary statistics, standard errors, and number of variant carries for PLA2G12A for all burden masks (with and without and conditioning on rs114816312 and rs41278045). We found that for the missense mask (Genebass: "Missense,LC" vs all masks in ST9) had similar exome-wide

significant p-values and directions of effect, but that the effect size estimate in Genebass was 10-fold lower. This difference in scale of effect sizes is most likely from log normalization we performed that was not done in the Genebass analysis. Furthermore, in our analysis, removal or conditioning on rs114816312 and rs41278045 still results in an exome-wide significant association supporting that the signal is not driven by these two variants alone.

Thus we conclude that our burden tests and the genebass resource show that loss of function in PLA2G12A confers increased TG/HDL. The difference in association statistics likely results from unit normalization (we performed log normalization of TG/HDL, Genebass did not) as well as software tools and data releases of the UKbiobank exomes. The consistency of the findings with using different methods and software demonstrates robustness of the results. As the reviewer requested we have added substantial detail to the Methods regarding QC and processing of the individual level data from the UK biobank exomes (lines 710-731).

7) In general, increased detail in the methods would aid ease of readability and reproducibility.

This is particularly the case in the following sections, in addition to those mentioned above:

7a. Line 535-536 – Please briefly describe the GLGC guidelines that were used to filter the samples. Based on the referred guidelines, clarification as to what data fields were used to determine lipid lowering medication use in UK Biobank is also needed.

7b. Line 546 – Please specify what reference panel was used for imputation.

7c. Lines 565-572 – a brief description of the Mass General Biobank would be helpful (e.g., recruitment, age, etc), as has been done for UK Biobank.

7d. RNA-sequencing of tissue samples, lines 680-701. More detail is needed to give a clearer view on this analysis as well as an overview of sample processing and data generation e.g., RNA extraction, sequencing, data QC, processing, and statistical analysis. Whilst microarray expression analysis is mentioned in reference 84 (Sears, 2009; PNAS – also reference 16) RNA sequencing is not mentioned. This is also the case for SGBS cells at lines 704-709.

We thank the reviewer for pointing out these areas.

7a) The following additional details have been added to the methods section describing the GLGC guidelines used to filter the samples: “To remove potential outliers and individuals with poor data quality, we applied several sample level filters to the UK Biobank participants following the published standards. Specifically, we retained only those individuals with a sample call rate greater than 95% and with heterozygosity values less than the median + 3x the interquartile range (IQR), and excluded those with reported gender-genetic sex mismatch.”

Notably, we did not adjust for cholesterol lowering medications as according to GLGC, non HDL-cholesterol measurements (total cholesterol and LDL cholesterol) levels are adjusted to account for cholesterol lowering medications, but not triglycerides or HDL-cholesterol.

7b) The following was added to the methods: “For imputation, the Haplotype Reference Consortium (HRC) and UK10K / 1000 Genomes reference panels were used as described in Bycroft et al.”

7c) The following additional information about the MGBB has been added into the methods: “The MGBB is an enterprise-wide biobank which has enrolled patients aged 20-99 from across Mass General Brigham, a healthcare system located primarily in the Boston area of the United States(Boutin et al. 2022).”

7d) Additional details regarding the mRNA sequencing methods and analysis of samples (muscle and adipose) from the Sears 2009 study as well as the SGBS cells during differentiation are provided (lines 771-809).

8) In general, more explicit disclosure throughout of what results were obtained from novel analyses compared previously published publicly available datasets would be helpful. Sometimes this is hard to deduce in the current manuscript. An additional section in the methods outlining the exact data sources used would be helpful for reproducibility.

As suggested by the reviewer, an external data sources section has been added with links to external data sources for clarity and reproducibility.

Minor comments:

9) Pearson’s correlation is stated to have been used in line 81 whereas Spearman’s correlation is stated in SF1 (p47) and methods (L547), is this a typo? Please clarify the approach taken in the text.

We thank the reviewer for pointing this out. Spearman rank correlation was the test used and we have corrected this in the results.

10) Please highlight which of the 251 loci identified are novel for insulin resistance, and other glycemic traits.

Novel TG/HDL loci for insulin resistance have been highlighted in ST4 (originally ST3) under column label “novelty”.

11) Sex specific GWAS analyses focus on examining the sex dimorphic effect of the 251 loci identified for TG/HDL in sex combined analyses, using the same analytical approach. Whilst the comparison of these loci is included, it would be informative describe these results more fully. Particularly to state if any additional genome wide significant loci were identified in the sex stratified analyses on top of those identified in the sex combined GWAS.

We appreciate this point and for completeness have added ST2 to detail 15 genome-wide significant loci identified in the sex-stratified GWAS. Overall we did not focus on these loci as they did not have a sufficiently high TG/HDL “boost” ($TG/HDL_{BS} > 1.5$) or had been previously identified in other studies.

12) Alongside the measures of fat distribution (e.g. VAT adjBMI), it may also be helpful to include BMI and WHR unadjusted for BMI allow a comparison of effects on overall adiposity with

fat distribution using look ups of the BMI adjusted results for the MRI-based measures included in Agrawal et al (2022). Indicating more clearly what summary statistics were obtained from BMI adjusted GWAS analyses (e.g. WHR, ASAT) would aid readability (Figure 5, lines 350, 355-366)

We thank the reviewer for suggesting this clarification and have added text to the results and Figure 5 clarifying that adiposity measurements shown were adjusted for BMI. Additionally, we extracted the BMI UNadjusted results and have incorporated them into ST11 (originally ST7) to facilitate comparison with the BMI adjusted values. We observe that strength of association increases upon adjustment for BMI.

13) Supplementary table 3 and Figure 2: In Supplementary Table 3 the SNP, beta, se and p value provided for look up of Fasting insulin and HOMA-IR are combined. Please split these into individual traits so it is clear what trait the summary statistics for each variant are referring to. Please define what is classed as no_overlap. Does this refer to those that do not meet genome wide significance threshold of $P < 5e-8$? It would be useful to see which loci have consistent expected direction of effects at a more relaxed significance threshold than that included at present.

Per the reviewer's suggestion we have updated ST4 (originally ST3) with association statistics for fasting insulin (FI) and HOMA-IR separately. We initially aggregated HOMA-IR and FI as a single phenotype group for visualization as the HOMA-IR summary statistics were derived from a much older study (PMID: 20081858) with fewer samples ($n=37,037$) and utilized a less stringent significance threshold ($P\text{-value} < 2 \times 10^{-5}$). Thus, to harmonize with more modern studies with larger sample sizes we required that all significant HOMA-IR ($p < 2 \times 10^{-5}$) loci also meet the commonly accepted ($p < 5 \times 10^{-8}$) for genome-wide significance in at least one other study to be included in analysis.

"No_overlap" in ST4 (originally ST3) was listed if significant SNPs from the previous insulin resistance related GWAS ($p < 2 \times 10^{-5}$ for HOMA-IR or $p < 5 \times 10^{-8}$ for all other traits) did not fall within the locus boundaries of a TG/HDL associated locus identified in our analysis. This description has been added into ST4 for additional clarification.

Reviewer #2 (Remarks to the Author):

In this manuscript the authors used the UK biobank dataset ($N= 382,129$) to discover loci associated with TG/HDL ratio, a marker of insulin resistance. The authors found 251 independent loci, of which 62 showed stronger association with TG/HDL ratio than with either of the measures, TG or HDL. Using additional fine-mapping approaches and other datasets, the authors go onto to describe some loci in more detail.

This is a really nice piece of work which is clearly written and easy to follow.

Major comments:

1) Given the just published paper but Oliveri et al., 2024, it would be nice to understand a bit better how the present results compare with those of the Oliveri et al. especially seeing as the underlying data are the same. Both this study and the Oliveri et al 2024 use the same TG/HDL ratio and the same discovery dataset (UK biobank). The sample sizes and software packages used differ, and the number of resulting independent loci also differ. It would be helpful (and reassuring to the reader) to try and find out the overlap in the findings and the level of concordance in the results.

We agree with the reviewer that given the publication of Oliveri et. al. 2024, it is important to evaluate the concordance of association signals and highlight the differences to enable the reader to contextualize both studies. We have performed a detailed comparison of both studies which are now described in the discussion. Of the 251 genome-wide significant TG/HDL loci identified in our study 241(96%) were matched either directly (n=222) or through proxy SNPs (n=19) in the Oliveri et al. 2024 published summary statistics. Despite differences in methodology and sample size, the correlation between the effect sizes for these 241 loci was 0.986 and the correlation of p-values 0.985 indicating high concordance. 207/241 of our identified loci had $p < 5 \times 10^{-8}$ in the Oliveri analysis with most of the remaining just above this threshold. Of the remaining loci (n=10) that could not be matched with Oliveri summary statistics, 9 were on the X chromosome (included in our GWAS but not the Oliveri GWAS) and one was very rare (MAF = 0.001).

Oliveri et al. report 369 significant loci of which we were able to identify 368 in our analysis (n=222 direct overlap lead SNPs and n=146 via proxy SNPs). The correlation between the effect sizes for these 368 loci was 0.978 and the correlation of p-values 0.966 indicating high concordance. 273/368 of the loci identified in Oliveri had $p < 5 \times 10^{-8}$ in our analysis with most of the remaining just above this threshold.

In contrast to Oliveri, our study includes the X chromosome, indel variants, and low frequency variants (MAF \geq 0.001 vs Oliveri MAF \geq 0.01) with scientifically important ramifications on genetic discovery. For example, at the PLA2G6 locus, the lead variant identified in our study is a deletion (rs200725415, 22:38575498:CT:C) has the highest PIP for being a causal variant in fine mapping analysis (ST6, originally ST4). While this locus is also identified in the Oliveri GWAS via the proxy SNP (rs2267373, 22:38600542:C:T) it does not meet the threshold (PIP > 0.1) for a causal variant and the more likely causal variant identified by our study is filtered out in Oliveri study because it is a deletion. Similarly, at the PLA2G12A locus the higher MAF threshold employed by Oliveri filters out causal coding SNPs (rs114816312, 4:110638824:C:T) in the locus identified in our study. In summary, the two studies share a high level of locus level concordance with some important design differences that result in the identification of some unique variants and loci.

2) This present work has a different emphasis in the post-GWAS analysis, so I don't believe the other publication detracts from the findings described here. I just feel that given that the two discovery analyses are so similar it would help the reader to better understand the differences and also the consistency, or lack thereof, of the overall findings.

We appreciate the reviewer's comment and agree regarding the scientific value of contextualizing both studies. In the previous question we detailed large overall consistency between loci identified in both studies with some key differences in findings due to our inclusion of indel variants, variants on the X chromosome and low frequency variants. Here we highlight the detailed methodological differences that result in different numbers of analyzable samples ($n = 402,398$ (Oliveri) v.s. $n = 382,129$ in our study) which could impact significance testing.

The Oliveri study selected participants who self-identified as "white British" and included an additional 52,702 participants categorized as European ancestry by genotyping data. Participants were excluded if they were missing information in any of the following categories: TGs, HDL, sex, age, and genetic principal components of ancestry (PC1-10). Variants were excluded from analysis if they had an imputation cutoff < 0.85 or minor allele count < 3.5 . Analysis was performed using SAIGE.

Our study did not select participants based on ancestry and followed standard protocols (PMID: 34887591) for participant, sample, and SNP level filtering (diagrammed in new SF1B). We did exclude individuals with missing information in any of the following categories: TGs, HDL, sex, age, and genetic principal components of ancestry (PC1-20). At the variant level we excluded SNPs with call rate $< 98\%$, HWE $p < 5 \times 10^{-8}$, imputation cutoff < 0.4 (i.e. INFO score), and minor allele count < 764 (corresponding to $MAF < 0.001$). We performed significant sample level filtering retaining only individuals with a sample call rate greater than 95% and with heterozygosity values less than the median + 3x the interquartile range (IQR), and excluded those with reported gender-genetic sex mismatch. Analysis was performed using REGENIE.

Overall, we believe that the sample size difference in the two studies are likely due to additional sample level filters we deployed to ensure high quality samples for genetic analysis.

3) I found it particularly perplexing that some of the top novel loci discussed and analysed in detail here, I could not see in the supplementary tables of Oliveri et al. e.g. PLA2G12A, PLA2G6 and VLL. It could be that the index variants were different, or that I missed them but it would be good to understand the key similarities and differences in the results obtained.

We thank the reviewer for pointing this out. These loci (PLA2G12A, PLA2G6, and VGLL3) highlight methodological differences in our two studies that cause some novel loci to be absent from the top associations highlighted in Oliveri.

1. PLA2G12A: the lead variant (rs114816312, $MAF = 0.008$) PLA2G12A reaches genome-wide significance in both studies from review of the full summary statistics, but is filtered out of the Oliveri study because it falls below their chosen frequency threshold ($MAF \geq 0.01$)
2. PLA2G6: The lead variant of PLA2G6 is a deletion (rs200725415, 22:38575498:CT:C) so is filtered out of Oliveri, but both studies identify this signal as a close proxy SNP (rs2267373; $r^2 = 0.94$), is among the top loci listed in Oliveri.
3. VGLL3: the lead variant in our study exceeded the threshold for genome-wide significance in our analysis ($p = 1.98 \times 10^{-11}$) but did not in the Oliveri analysis ($p_{\text{Oliveri}} = 1.7 \times 10^{-6}$). Nevertheless, it strongly exceeded the nominal threshold of significance in

both studies. This is possibly due to differences in sample sizes, filtering and software tools utilized in our two studies.

4) When comparing the genetic correlations of TG/HDL ratio with some other measures and diseases, the authors compare the genetic correlation of TG/HDL with that of other glycaemic traits with the same traits. It wasn't always clear to me why some statements were made, as the measures (without a statistic to test for differences) looked broadly similar by eye. For example, the authors state "Positive genetic correlations were observed between TG/HDL and T2D ($r_g=0.51$, $p=1.9 \times 10^{-36}$), CVD ($r_g=0.31$, $p=2.8 \times 10^{-28}$), and NAFLD ($r_g=0.74$, $p=0.001$). Notably, TG/HDL had comparable genetic correlation with T2D as compared to glycaemic traits (FI $r_g=0.6$, $p=5.2 \times 10^{-29}$; FG 131 $r_g=0.57$, $p=3.8 \times 10^{-18}$) but greater genetic correlation with CVD (FI $r_g=0.29$, $p=3 \times 10^{-9}$; FG $r_g=0.12$, $p=1 \times 10^{-4}$) and NAFLD (FI $r_g=0.9$, $p=0.01$; FG $r_g=0.3$, $p > 0.05$)." So the authors claim that the genetic correlation of TG/HDL with T2D $r_g=0.51$ is comparable to that of glycaemic traits (FI $r_g=0.6$; FG $r_g=0.57$); but then claim they see "greater genetic correlation with CVD". However, their data show TG/HDL $r_g=0.31$ with CVD vs $r_g=0.29$ for FI and CVD. I'd say this is pretty comparable too? Same for NAFLD TG/HDL $r_g=0.74$, while FI is $r_g=0.9$? In fact, for the latter FI looks more correlated with NAFLD.

The reviewer's point is well taken. We have tempered the language and removed this statement about "greater" genetic correlation with metabolic disease outcomes. In place we further describe the genetic correlation between TG/HDL and other surrogate measures of insulin resistance, glycaemic traits, and beta cell function (ST3, originally ST2): "When considering glycaemic traits and insulin secretion, the magnitude of genetic correlation between TG/HDL and HbA1c ($r_g=0.12$) or 2hr glucose ($r_g=0.23$) are comparable to FG ($r_g=0.17$) and smaller than genetic correlation between TG/HDL and FI ($r_g=0.49$), HOMA-IR ($r_g=0.49$) or ISI ($r_g=-0.47$). Even though HOMA-B and HOMA-IR are calculated from the same fasting measurements, HOMA-IR ($r_g=0.49$, $p=4.11 \times 10^{-10}$) had a stronger and more significant correlation with TG/HDL than HOMA-B ($r_g=0.41$ $p=4.78 \times 10^{-7}$). Overall, these analyses show that TG/HDL is more genetically correlated with measures of insulin resistance than glycemia or insulin secretion."

5) For the boosted score the authors compute, it would be helpful to be clear in the methods whether the $-\log_{10}$ p-value for all three traits being compared TG/HDL, TG and HDL came from association testing of the exact same samples so there are no underlying differences in sample sizes between the traits.

We appreciate the suggestion. The same number of UK Biobank participants ($n=382,129$) was used for TG/HDL, TG and HDL. We have additionally modified the text in the results section for more clarification: "Using the lead SNP of each TG/HDL risk locus, we computed a "boost score" (TG/HDL_{BS}) utilizing the association p-values for TG/HDL, TG, and HDL for that SNP in **the UK Biobank (computed from the same samples with identical QC and analysis; Methods).**"

6) In the fine-mapping section it would be helpful to know what made loci not mappable?

We apologize for the confusion. We use the term “mappable” (57/62 loci) to indicate where the SuSie algorithm converged to produce at least one credible set containing at least one SNP. We have modified the text of the results as follows: “Across the 57 mappable loci (**i.e. algorithm converged to produce at least one credible set with one SNP**), the defined credible sets contained a median of 66 variants...”

7) For the nominated gene PLA2G12A, the authors show the variant rs114816312 has $\beta=0.1$, this particular variant is predicted to lead to a missense, p.D111N. A second variant, rs41278045 is also predicted to lead to a missense, p.C131R, and both are predicted to be deleterious. Still, when performing gene-burden tests with predicted LOF variants the effect size is one order of magnitude smaller 0.01 (mentioned in the text and shown in ST5) but Figure 4B shows a different effect size. Do the authors have a proposed mechanism for how missense variants will affect the TG/HDL ratio a lot more than a burden of LOF variants, when the authors propose that in both situations the missense variants are impacting protein function? Another question I have here is whether the burden of LOF effects changes when conditioned on rs114816312 and rs41278045 variants? The authors removed the two variants (rs114816312 and rs41278045) from the burden but if there is LD between either of them and some of the other variants in the burden could this be the reason for the burden effects shown?

We thank the reviewer for noticing this discrepancy in effect sizes from the SNP genotyping association analysis, our own exome burden tests, and the Genebass lookup burden tests at the PLA2G12A locus. We share the reviewer’s view that a gene-burden test with predicted LOF variants should have an effect size similar to or greater than a single predicted missense LOF variant although depending on allele frequency the gene-burden tests may have less statistical power. In short, we think the discrepancy in effect sizes is likely due to log normalization performed in our study on TG/HDL values (also performed by Oliveri et al.). Supportively, our genotyping based association effect size for rs114816312 ($\beta = 0.14$) concurs well with Oliveri et al. ($\beta_{\text{Oliveri}} = 0.19$) and with the effect size of the other missense PLA2G12A SNP rs41278045 ($\beta = 0.18$, ST6). These effect sizes are in the range of what we identify in gene-burden tests across various variant aggregation masks for deleterious variants in PLAG12A ($\beta = 0.15 - 0.19$, new ST9) where TG/HDL ratio was also log normalized prior to analysis. Finally, we examined PLA2G12A:TG/HDL association from the missense mask (Genebass: “Missense,LC”: $p = 3.28e-48$) and compared to our own missense burden analyses (ST9) finding similar exome-wide significant p-values and directions but a 10-fold lower effect size estimate in Genebass. We have clarified this in the text and ST5 (reindexed to ST7).

We appreciate the reviewers point considering the independence of the burden signal from rs114816312 and rs41278045. Formally conditioning on rs114816312 and rs41278045 does not change the gene-burden estimates (ST9) indicating that the burden signal does not depend on these variants. Furthermore rs114816312 and rs41278045 are independent of one another ($r^2=6.34e-6$). This has been incorporated into the results (lines 266-274).

8) At the TNFAIP8 locus the authors refer to serum proteomics data from a different study, have the authors looked directly at the proteomics data available in biobank?

Thank you for this fantastic suggestion! We investigated the plasma levels of TNFAIP8 in the Olink Explore 3072 proteomics data measured across 54,219 UK Biobank participants (PMID: 37794186) in combined and sex-stratified analysis finding that TNFAIP8 serum levels correlated with TG/HDL, HbA1c and WHR in both sexes but with much stronger association in females! We have summarized these findings in a new ST10 and added them to the results (lines 363-368). Again we thank the reviewer for this suggestion that has greatly strengthened our findings at the TNFAIP8 locus.

9) Overall, in terms of nominating causal genes, I think the authors need to express a little more caution as the data shown are not definitive. Namely, the authors focused their analysis on variants that had the higher PIP in their credible set but in some examples this was still not particularly strong (for example, PIP=0.1 in PLA2G6 and also see concerns about the LOF burden at PLA2G12A), and the authors have not shown that those specific variants definitely affect protein function/expression in an allele-specific manner. Nor have they shown a direct link between protein function and the measure they are looking at (TG/HDL ratio). So there is still room for other variants (and genes) at these loci to be implicated. I'd suggest that they propose these as candidate causal as they have shown correlative data, but not mechanistic data direct linking specific variants to gene expression or protein expression/function.

We agree with the reviewer and have tempered "causal gene" to "candidate causal gene" in the revised manuscript.

10) I also think the authors need to be more cautious regarding nomenclature as their "causal variants", should more correctly be called "putative causal variants". For most of the results shown, the 95CS still had multiple variants, and the authors focused on those with highest PIP values (and as mentioned above PIP=0.1 is not that high) leaving room for other variants to be causal, or also causal. In addition, any statistical fine-mapping approach is limited by the SNPs present in the data (the true causal variant may not actually be in there but just others tagging it) so it is not possible to be certain the best PIP variants are indeed causal, especially in the absence of functional data directly modelling effect of that variant.

We thank the reviewer for this suggestion and have updated "causal variants" to "putative causal variants" in the revised manuscript.

11) The below is a minor point but in the discussion the authors cannot claim to have the largest GWAS for TG/HDL ratio given the recently published paper which actually had a marginally larger sample size of 402,398. There are few places in the discussion where this now needs to be updated and probably refer to this other published paper at least by stating while this paper was under review this other one was published?

Agreed, we have removed reference to the “largest” GWAS for TG/HDL and included a section in the discussion contextualizing our findings with those published by Oliveri et al. (lines 476-498)

Reviewer #3 (Remarks to the Author):

The rationale of this paper is questionable as this reviewer was not aware of a way to measure insulin sensitivity in general populations using lipids values. Reading the ref 15 paper (only paper referred to justify this study) it is said : “The optimal triglyceride/high-density lipoprotein cholesterol ratio for predicting insulin resistance and LDL phenotype was 3.5 mg/dl; a value that identified insulin-resistant patients with a sensitivity and specificity comparable to the criteria currently proposed to diagnose the metabolic syndrome. The sensitivity and specificity were even greater for identification of patients with small, dense, LDL particles. In conclusion, a plasma triglyceride/high-density lipoprotein cholesterol concentration ratio $>$ or $=3.5$ provides a simple means of identifying insulin-resistant, dyslipidemic patients who are likely to be at increased risk of cardiovascular disease.” It is obvious that this old study (never reproduced) was intended to determine most insulin resistant subjects among high risk of CV diseases people, with a threshold for IR, but not to have a continuous measurement of insulin sensitivity in general population. Therefore this reviewer expressed concern that all the current study is flawed. This feeling is reinforced by the so called “replication” from an old study (ref 16, with only 45 subjects, which did not show this correlation). The supp figure 1 clearly shows that at medium (most frequent in general population) Trig/HDL ratio the clamp measures insulin sensitivity is extremely variable, showing again that this analysis is probably valuable for a fragment of at risk for CV patients but not as an epidemiology tool.

The GWAS analysis for Trig/HDL ratio is standard, with no obvious replication in a medium size American cohort. The correlation with previous GWAS for T2D etc was expected due to the abnormal lipid values in these insulin resistant populations.

We appreciate the reviewer’s concerns regarding using the TG/HDL ratio as a surrogate marker for insulin resistance and apologize for citing only one study despite there being many. The study we cited (McLaughlin 2005) suggested TG/HDL ≥ 3.5 as a potential clinical cutoff to identify insulin resistant individuals based on receiver-operator curve analysis to maximize specificity. But this study also assessed the pairwise continuous relationships between fasting insulin, TG/HDL and steady state glucose from insulin suppression tests finding that TG/HDL correlated as strongly with steady state glucose as did fasting insulin (McLaughlin 2005; Table 1). Other studies have also found a continuous relationship between insulin resistance and TG/HDL in population sized cohorts across different ethnicities (PMID: 34678755, PMID: 22004541), weight categories (PMID: 30641729, PMID: 37582739), and in children (PMID: 37582739). Some of these studies suggest different TG/HDL cutoffs for the purpose of clinically classifying individuals, but they all observe a quantitative relationship. To the reviewer’s point the continuous relationship between insulin resistance and TG/HDL is potentially non-linear across the entire range of values, but maintains linearity around TG/HDL < 1 which captures the most frequent values in the general population (PMID: 34678755; Figure 2).

Furthermore as we show in our manuscript and others have recently published as well (PMID: 38200128) that genetic association performed with TG/HDL as a continuous trait identifies many known insulin resistance genes (e.g. IRS-1, PPARG) and TG/HDL has a strongly shared genetic basis with other surrogate markers of insulin resistance.

We thank the reviewer for pointing this out and have bolstered the rationale by incorporating these additional references into the introduction of the revised manuscript.

This reviewer is not convinced that the “boosted priority TG/HDL association study for specific IR genes” has any physiology meaning, other than being a statistical game. The “boost score” is for this reviewer artificial and in this reviewer’s memory LPL is not a T2D associated gene.

We utilized the “boost score” as a heuristic to rank TG/HDL associated loci according to likelihood of being insulin resistance loci and show empirically that the score ranks known insulin resistance genes including IRS1 and PPARG above known lipid genes such as CETP and ANGPTL3. This was used as one of many methods to prioritize genes for follow up analysis (Figure 3). We do not intend to claim that it has any physiological meaning and explicitly describe the limitations of the boost score in the discussion: “prioritized “boosted” genes that showed stronger association with the TG/HDL ratio than with TG or HDL alone, but we cannot exclude the possibility that some of these genes are also involved in lipid metabolism (e.g. APOB, TG/HDLBS = 5.5).”

We apologize for any confusion. We report LPL as an insulin resistance gene, NOT a T2D associated gene. The study we cite takes a human genetic approach to linking LPL haplotypes to glucose infusion rates (PMID: 14693718), but experimental evidence also shows that LPL misexpression in tissues can cause tissue specific insulin resistance (PMID: 11390966). We have added this second study to further bolster the references that LPL is an insulin resistance gene.

The search for “causative” genes is interesting but it is apparently only based on silico data from databases and previous papers with no novel and original mechanistic proof of concept that any of the “new” genes is physiologically associated with both insulin sensitivity and lipids metabolism. BTW the conclusion of the paper shows what remains to be done, from the current data, enrich our knowledge of the physiology of the crosstalk between lipid and glucose metabolism.

We appreciate the reviewer’s interest in the novel genes highlighted and agree that the next step scientifically would be to undertake further mechanistic work including in vivo modeling to understand how these genes control insulin sensitivity at a molecular and physiological level. Such studies are beyond the scope of this manuscript, but we are excited to pursue and report on these in future work.

REVIEWERS' COMMENTS

Reviewer #1 (Remarks to the Author):

In the revised manuscript DeForest and colleagues respond in detail to most of the reviewers' comments, including providing additional analyses as appropriate. As a reviewer of the original submission, I thank the authors for their thorough responses. Greater detail provided in the methods in the resubmission has aided understanding of the analyses. Further, the inclusion of comparisons of the results and analytical approach reported in the recently published paper by Oliveri et al (Nature Genetics) aids direct comparison of results between these studies. The response to R2 highlights that analytical and filtering differences explain the lack of reporting of PLA2G12A, PLA2G6 and VGLL3 in Oliveri et al despite being identified and subsequently prioritised in this manuscript.

The concerns about the measure of TG/HDL ratio being a specific enough surrogate marker of insulin resistance remain. The possibility of TG/HDL capturing genetic loci implicated in other pathways (e.g. lipid metabolism), not related to insulin resistance, could be highlighted more clearly throughout the manuscript, ahead of the concluding paragraph where this is currently predominantly discussed. This caveat of using a proxy measure, that likely captures other pathways including lipid metabolism, should also be considered in the reporting of the number of 'novel' loci for insulin resistance identified and their comparison of overlap with other traits. At lines L181-183 the number of loci not previously implicated in insulin resistance or metabolic disease is compared with the full 251 loci, despite earlier in the paragraph the authors highlighting that only 43% of these loci had a 'boost score' that met the threshold defined to distinguish them from loci for triglycerides and/or HDL alone. A more direct comparison could be the 109 loci that had a positive boost score or those within the top quartile of boost scores which are prioritised.

In the comparison with Oliveri et al included in the resubmission (reply to R2 major comment 3), the authors highlight the inconsistency in the p-value of the lead variant at VGLL3 in Oliveri et al compared to this manuscript (p-value DeForest = 1.98×10^{-11} , p-value Oliveri = 1.7×10^{-6}). There was a lack of ancestry filtering in the GWAS conducted by the authors, which is one of the differences with the analysis by Oliveri et al which was restricted to European ancestry individuals. It would be interesting to see if potential ancestry-specific effects drive this difference.

I also include below some follow-up on specific responses to my previous comments (Reviewer 1), for points where concerns remain:

[R1 Major comment 1] Thank you for clarifying the TG/HDL ratio was calculated in the fasting state for individuals with the glycaemic clamp. This further highlights the concern about TG/HDL ratio being a suitable proxy. A one-to-one comparison of the ability of TG/HDL to proxy insulin resistance in this small-scale clamp study and UK Biobank cannot be drawn with the measures in UK Biobank being in a non-fasting state. It remains important to understand how transferrable the TG/HDL ratio is to non-fasted samples as a proxy specifically for insulin resistance. Whilst I appreciate it may not be possible to directly test in the data sources available, this caveat should be discussed in the manuscript.

[R1 Major comment 4] The reported genetic correlation of non-fasted TG/HDL and the observational correlations of fasted TG/HDL with glycaemic traits are approximately comparable. However, these correlations are very modest for a trait that is being used as a proxy for insulin resistance. These correlations are also significantly weaker than the correlations of TG/HDL with triglyceride and HDL measures alone (from ST3: FI $r_g = 0.49$, TG $r_g = 0.96$ and HDL $r_g = -0.82$). This weak correlation highlights the caveat of using TG/HDL as a proxy of insulin resistance. As discussed by the authors, TG/HDL likely captures other biological processes related to triglycerides and/or HDL, as well as insulin resistance. Therefore, not all the genetic loci identified are likely to be associated with insulin resistance and therefore there are likely several false positive loci reported for insulin resistance using TG/HDL as a measure.

Reviewer #2 (Remarks to the Author):

The authors have addressed all of my previous concerns. I have no further comments.

Reviewer #3 (Remarks to the Author):

The reviewer thanks the authors for their responses. This reviewer does not discuss the quality of the statistical study but still questions the novelty and pathophysiology interest of this paper that only marginally improves the outcomes of the recent other GWAS paper on trig/HDL in the UKbiobank (it is mostly confirmatory) and in addition does not respond to the most fundamental question of GWAS of some "exotic" phenotypes (lipids are exotic with regards to glucose metabolism) that are more and less linked to insulin sensibility: what does this quantitative phenotype brings for the genetics of this key trait (insulin sensibility)? Comparing this paper results with those obtained in GWAS for HIP/waist ratio, INS values and clamps data would be interesting. BTW I don't think that this study truly characterizes insulin resistance pathways but rather explores lipid metabolism which can be impaired in insulin resistance states.

REVIEWERS' COMMENTS

Reviewer #1 (Remarks to the Author):

In the revised manuscript DeForest and colleagues respond in detail to most of the reviewers' comments, including providing additional analyses as appropriate. As a reviewer of the original submission, I thank the authors for their thorough responses. Greater detail provided in the methods in the resubmission has aided understanding of the analyses. Further, the inclusion of comparisons of the results and analytical approach reported in the recently published paper by Oliveri et al (Nature Genetics) aids direct comparison of results between these studies. The response to R2 highlights that analytical and filtering differences explain the lack of reporting of PLA2G12A, PLA2G6 and VGLL3 in Oliveri et al despite being identified and subsequently prioritised in this manuscript.

We are happy to have satisfied the reviewer's questions and thank the reviewer for the many suggestions that have clarified and strengthened the manuscript.

The concerns about the measure of TG/HDL ratio being a specific enough surrogate marker of insulin resistance remain. The possibility of TG/HDL capturing genetic loci implicated in other pathways (e.g. lipid metabolism), not related to insulin resistance, could be highlighted more clearly throughout the manuscript, ahead of the concluding paragraph where this is currently predominantly discussed. This caveat of using a proxy measure, that likely captures other pathways including lipid metabolism, should also be considered in the reporting of the number of 'novel' loci for insulin resistance identified and their comparison of overlap with other traits. At lines L181-183 the number of loci not previously implicated in insulin resistance or metabolic disease is compared with the full 251 loci, despite earlier in the paragraph the authors highlighting that only 43% of these loci had a 'boost score' that met the threshold defined to distinguish them from loci for triglycerides and/or HDL alone. A more direct comparison could be the 109 loci that had a positive boost score or those within the top quartile of boost scores which are prioritised.

Agreed, we have added the following sentence to the results highlighting the top quartile of boost scores: "Considering the subset of the 62 top-quartile boosted loci, 29 had not been previously identified."

In the comparison with Oliveri et al included in the resubmission (reply to R2 major comment 3), the authors highlight the inconsistency in the p-value of the lead variant at VGLL3 in Oliveri et al compared to this manuscript (p-value DeForest = 1.98

10^{-11} , p -value Oliveri = 1.7×10^{-6}). There was a lack of ancestry filtering in the GWAS conducted by the authors, which is one of the differences with the analysis by Oliveri et al which was restricted to European ancestry individuals. It would be interesting to see if potential ancestry-specific effects drive this difference.

The lead variant for VGLL3 (rs13066793) using our analysis pipeline in European only samples still had a genome-wide significant association ($p = 4.71 \times 10^{-10}$). This suggests that the differences between our estimate and the Oliveri estimate is unlikely to be due to an ancestry specific effect.

I also include below some follow-up on specific responses to my previous comments (Reviewer 1), for points where concerns remain:

[R1 Major comment 1] Thank you for clarifying the TG/HDL ratio was calculated in the fasting state for individuals with the glycaemic clamp. This further highlights the concern about TG/HDL ratio being a suitable proxy. A one-to-one comparison of the ability of TG/HDL to proxy insulin resistance in this small-scale clamp study and UK Biobank cannot be drawn with the measures in UK Biobank being in a non-fasting state. It remains important to understand how transferrable the TG/HDL ratio is to non-fasted samples as a proxy specifically for insulin resistance. Whilst I appreciate it may not be possible to directly test in the data sources available, this caveat should be discussed in the manuscript.

Agreed, we have highlighted this point in the limitations section of the discussion adding the following text: "These genetic correlations are observed even though the UK Biobank TG/HDL ratio was computed from non-fasting measurements, which could confound the use of TG/HDL as a surrogate insulin resistance marker."

[R1 Major comment 4] The reported genetic correlation of non-fasted TG/HDL and the observational correlations of fasted TG/HDL with glycaemic traits are approximately comparable. However, these correlations are very modest for a trait that is being used as a proxy for insulin resistance. These correlations are also significantly weaker than the correlations of TG/HDL with triglyceride and HDL measures alone (from ST3: FI $rg = 0.49$, TG $rg = 0.96$ and HDL $rg = -0.82$). This weak correlation highlights the caveat of using TG/HDL as a proxy of insulin resistance. As discussed by the authors, TG/HDL likely captures other biological processes related to triglycerides and/or HDL, as well as insulin resistance. Therefore, not all the genetic loci identified are likely to be associated with insulin resistance and therefore there are likely several false positive loci reported for insulin resistance using TG/HDL as a measure.

We agree and have emphasized this point further in the limitations section:
“Moreover, the TG/HDL ratio is influenced by the individual levels of TG and HDL, which share loci with the TG/HDL ratio **and could result in true locus associations but false positives for insulin resistance.**”

Reviewer #2 (Remarks to the Author):

The authors have addressed all of my previous concerns. I have no further comments.

Reviewer #3 (Remarks to the Author):

The reviewer thanks the authors for their responses. This reviewer does not discuss the quality of the statistical study but still questions the novelty and pathophysiology interest of this paper that only marginally improves the outcomes of the recent other GWAS paper on trig/HDL in the UKbiobank (it is mostly confirmatory) and in addition does not respond to the most fundamental question of GWAS of some "exotic" phenotypes (lipids are exotic with regards to glucose metabolism) that are more and less linked to insulin sensibility: what does this quantitative phenotype brings for the genetics of this key trait (insulin sensibility)? Comparing this paper results with those obtained in GWAS for HIP/waist ratio, INS values and clamps data would be interesting. BTW I don't think that this study truly characterizes insulin resistance pathways but rather explores lipid metabolism which can be impaired in insulin resistance states.

We appreciate the reviewer's concerns. The comparisons with GWAS for waist-hip ratio (WHR) and fasting insulin (FI) that the reviewer flags as potentially interesting comparisons were also of high interest to us and are presented in Figure 2b alongside comparisons with other surrogate markers and metabolic disease GWAS. We found that out of 251 loci identified in our study WHR shared 93 overlapping associated loci with TG/HDL and FI shared 27 associated loci (Figure 2b). Notably, 118 out of 251 loci had not previously been implicated as insulin resistance or metabolic disease risk loci and did not overlap with any of the prior GWASs. These findings support that there is significant but incomplete overlap between the physiology and genetic architecture of TG/HDL and other surrogate insulin resistance markers, and that each study reveals novel loci/ genes.